# Odd surface waves in two-dimensional incompressible fluids

**Alexander G. Abanov[1,2], Tankut Can[4] and Sriram Ganeshan[3,1]**

**1** Simons Center for Geometry and Physics, Stony Brook University,
Stony Brook, NY 11794, USA
**2** Department of Physics and Astronomy, Stony Brook University,
Stony Brook, NY 11794, USA
**3** Department of Physics, City College, City University of New York,
New York, NY 10031, USA
**4** Initiative for the Theoretical Sciences, The Graduate Center,
CUNY, 10012, USA

## Abstract

We consider free surface dynamics of a two-dimensional incompressible fluid with odd viscosity. The odd viscosity is a peculiar part of the viscosity tensor which does not result in dissipation and is allowed when parity symmetry is broken. For the case of incompressible fluids, the odd viscosity manifests itself through the free surface (no stress) boundary conditions. We first find the free surface wave solutions of hydrodynamics in the linear approximation and study the dispersion of such waves. As expected, the surface waves are chiral and even exist in the absence of gravity and vanishing shear viscosity. In this limit, we derive effective nonlinear Hamiltonian equations for the surface dynamics, generalizing the linear solutions to the weakly nonlinear case. Within the small surface angle approximation, the equation of motion leads to a new class of nonlinear chiral dynamics governed by what we dub the *chiral* Burgers equation. The chiral Burgers equation is identical to the complex Burgers equation with imaginary viscosity and an additional analyticity requirement that enforces chirality. We present several exact solutions of the chiral Burgers equation. For generic multiple pole initial conditions, the system evolves to the formation of singularities in a finite time similar to the case of an ideal fluid without odd viscosity. We also obtain a periodic solution to the chiral Burgers corresponding to the non-linear generalization of small amplitude linear waves.

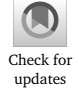

# 1   Introduction

Recently there has been much interest in the role of parity violating effects in two dimensional incompressible hydrodynamics. This was motivated by the seminal paper by Avron, Seiler, and Zograf [1] where they showed that the quantum Hall (QH) ground state has non-vanishing non-dissipative odd viscosity. In two dimensions, the odd part of the viscosity tensor is compatible with isotropy and has an elegant interpretation as the adiabatic curvature on the space of flat background metrics [1]. The role of odd viscosity in the context of QH fluids (where it is dubbed Hall viscosity) has been an active area of research [2–25].

Avron subsequently considered the case of classical 2D hydrodynamics with dominant odd viscosity [26]. Recent works have further outlined observable consequences of the odd viscosity in classical two dimensional incompressible hydrodynamics [27–31]. In three dimensions,

the general parity odd terms of the viscosity tensor were considered in the context of a plasma in a magnetic field [32] and in hydrodynamic theories of superfluid He-3A [33].

In this work, we consider the classical problem of deep water surface waves with the addition of odd viscosity. For simplicity, we refer to these as simply *odd surface waves*. The generalization of this work to the odd version of shallow water surface waves is straightforward. Our starting point is the incompressible 2D Navier-Stokes equation with free boundary and in the presence of odd viscosity [30]. These are basically waves on the 1D surface of a 2D fluid with broken parity symmetry. We parameterize the boundary of the fluid using a height function defined by $y = h(x, t)$ assuming that the fluid domain is defined by $y \leq h(x, t)$. We begin by solving the linearized (neglecting advective acceleration terms) 2D Navier-Stokes equation in the presence of odd viscosity ($\nu_o$) subject to no-stress boundary conditions. These linearized solutions are the ("odd") generalization of linearized viscous Lamb surface wave solutions (in the following just "Lamb's solutions") with shear viscosity ($\nu_e$) and non-zero vorticity [34]. The presence of odd viscosity completely changes the surface phenomenon and allows for chiral dispersing waves in the limit of no gravity ($g = 0$) and vanishing shear viscosity $\nu_e \to 0$. The dispersion relation in this limit is of the form

$$\Omega(k) = -2\nu_o k|k| \tag{1}$$

and is reminiscent of the famous Benjamin-Davis-Ono (BDO) dispersion relation.

Vorticity plays a significant role in the structure of linearized solutions in both Lamb's case and its odd viscosity generalization. In the limit of $\nu_e \to 0$, the vorticity is confined to a thin layer at the boundary. The thickness of this boundary scales as $\delta \sim \sqrt{\frac{\nu_e}{\nu_o}}$, similar to Lamb's case [34]. However, the presence of odd viscosity significantly alters the scaling of vorticity within the boundary layer, which diverges with vanishing shear viscosity ($\omega \sim \frac{1}{\sqrt{\nu_e}}$) as opposed to a constant vorticity $\omega \sim O(1)$ for Lamb's case. Outside of this layer the vorticity is negligible and the fluid can be approximated by an irrotational fluid which is completely determined by a scalar potential [35, 36]. Ruvinsky et al. [37] used this almost irrotational nature of Lamb's solutions with only shear viscosity and gravity and constructed a scheme dubbed the "quasi-potential approximation" (QPA) to capture the non-linear surface dynamics. The basic idea of QPA is to integrate out the vortical boundary layer and rewrite it as an effective Bernoulli's equation at the surface in terms of a scalar potential term. The integrated out vortical part modifies the boundary condition which determines the modified pressure at the boundary [38–40]. The boundary layer in incompressible dissipation free flows and the related non-analytic dispersion had been also discussed in the theory of edge modes in fractional Quantum Hall systems [41].

Following the QPA scheme of Ruvinsky et al., we obtain non-linear potential flow equations for a fluid with odd viscosity. As mentioned earlier, the presence of odd viscosity results in diverging vorticity $\omega \sim \frac{1}{\sqrt{\nu_e}}$ within the boundary layer compared to a constant vorticity in the analysis of Ref. [37] for shear viscosity. This altered scaling significantly changes the structure of the resulting non-linear equation compared to the one obtained by Ruvinsky et al. for the gravity waves. We derive one-dimensional Hamiltonian equations governing surface dynamics corrected by dispersive terms dependent on odd viscosity. This Hamiltonian dynamics within a small surface angle approximation [42] takes the form of the complex Burgers equation

$$u_t + 2uu_x - 2i\nu_o u_{xx} = 0, \tag{2}$$

where $u(x, t)$ is a complex function. The equation (2) is supplemented by an additional condition that $u(z, t)$ is analytic in the lower half plane of $z$. This analyticity condition enforces chiral dynamics at the surface and selects only special class of solutions from the ones of the simple complex Burgers case. Hence, we dub the complex Burgers with the analyticity condition as *chiral Burgers* equation from hereon.

The above equation can be transformed to Schrödinger equation $i\Psi_t = 2\nu_o\Psi_{xx}$ using Cole-Hopf transformation $u = 2i\nu_o\Psi_x/\Psi$, where we can identify $\nu_o = -\hbar/4m$. Note that this is chiral Schrödinger equation since $\Psi$ satisfies the analyticity condition similar to $u$ which makes the dynamics chiral. This is analogous to the transformation that connects Burgers equation to the diffusion equation.

The solution of this equation defines the small angle approximation to the non-linear velocity profile of the fluid. We analyze the dynamics and present some exact solutions of this effective non-linear equation. The chiral Burgers equation without the dispersive term (inviscid Burgers) has been previously obtained for surface waves in Ref. [42]. The odd viscosity adds the dispersive term and changes the character of formation of some of the singularities studied in [42].

This paper is organized as follows. We begin by introducing the general hydrodynamic equations in Sec 2 and derive the odd viscosity generalization of Lamb's solutions in Sec 4. We analyze these solutions in the limit $\nu_e \to 0$ and obtain the scaling of the boundary layer thickness and velocity and vorticity profile with respect to $\nu_e$ (Sec 4.2). Using this scaling for the linearized solutions, we derive the effective equation (chiral Burgers) governing the non-linear surface dynamics of the fluid with odd viscosity in Sec. 5. In section 6, we derive the non-linear Hamiltonian structure to the second order and deduce mass and momentum conservation laws. We obtain some exact solutions of this new non-linear equation and end the paper with future directions (Sec 7). Various technical results supporting the main text are presented in appendices. In particular, we also discuss a one parameter family of non-linear equations (Eq. 149), which contains the chiral Burgers equation and the Benjamin-Davis-Ono (BDO) equation as limiting cases.

# 2 Statement of the problem

## 2.1 Hydrodynamic equations

Let us begin from the main hydrodynamic equations with free boundary and in the presence of odd viscosity. In this paper we assume that temperature does not play any major role and the fluid is incompressible. Moreover, we assume that the density of the fluid is constant and take it to be unity $\rho = 1$. Under these assumptions, the equations of hydrodynamics can be written as:

$$\partial_i v_i = 0, \tag{3}$$
$$D_t v_i = \partial_j T_{ij} - \partial_i(gy). \tag{4}$$

Here the summation over repeated indices is assumed ($i, j = 1, 2$) and $D_t \equiv \partial_t + v_i \partial_i$ is a material time derivative. The first equation is the incompressibility condition. The potential $gy$ is an external gravitational potential. To have a closed system of equations one needs to define a constitutive relation expressing the stress tensor $T_{ij}$ in terms of the velocity of the fluid. We write:

$$T_{ij} = -p\delta_{ij} + \nu_e(\partial_i v_j + \partial_j v_i) + \nu_o(\partial_i^* v_j + \partial_i v_j^*). \tag{5}$$

The first term of the stress tensor (5) is the standard isotropic pressure term. The other terms come from the viscosity tensor. The second term takes into account the shear viscosity of the fluid with the coefficient $\nu_e$ known as kinematic shear viscosity. The last term in (5) is less familiar. This is the odd viscosity term. The effects of this term on surface waves are the main subject of this paper. The coefficient $\nu_o$ is known as kinematic odd viscosity (or Hall viscosity).

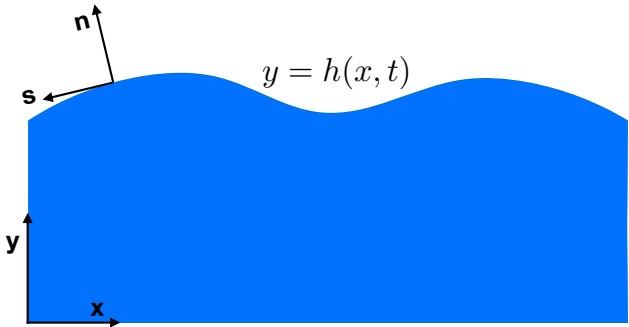

Figure 1: Fluid domain with free surface $y = h(x, t)$

In writing this term we introduced the notation

$$a_i^* \equiv \epsilon_{ij} a_j, \tag{6}$$

so that the "starred" vector $\mathbf{a}^*$ is just a vector $\mathbf{a}$ rotated by 90 degrees clockwise.

The following three important remarks are in order. 1) All equations we wrote so far are legitimate in any spatial dimension except for the odd viscosity term of (5) which is specific to two-dimensional hydrodynamics. It breaks parity (explicitly uses $\epsilon_{ij}$) without breaking isotropy of the two-dimensional fluid [26]; 2) The symmetric stress tensor (5) does not have the most general form for two-dimensional isotropic fluids with broken parity. Within this order in velocity gradients for the incompressible fluids, one more term can be added. It is the contribution to the diagonal part of the stress proportional to $\omega \delta_{ij}$, where $\omega = \boldsymbol{\nabla} \times \mathbf{v} = \partial_i v_i^*$ is the vorticity of the fluid (pseudoscalar in two dimensions). This term can be easily included into the discussion but is omitted for simplicity [1] ; 3) For an incompressible fluid the equation of state $p(\rho)$ is not needed. The pressure $p$ in this case is not a "state variable" as it is fully determined by the fluid flow.

To elaborate on the third remark it is convenient to rewrite (4) as

$$D_t v_i = -\partial_i(p - \nu_o \omega + g y) + \nu_e \Delta v_i, \tag{7}$$

where we used the expression (5) and the incompressibility condition (3). Taking a curl of this equation we obtain the vorticity transport equation

$$D_t \omega = \nu_e \Delta \omega. \tag{8}$$

In two dimensions, the two equations (3,8) define the two-component velocity field and one can find pressure $p$ from the known velocity field using (7). Notice that both equations (3,8) do not depend directly on odd viscosity $\nu_o$. This means that the effects of odd viscosity on the flow of an incompressible fluid can come only through boundary conditions [30].

## 2.2 Boundary conditions

Let us consider the fluid dynamics on a domain with boundary. In this case in addition to equations (3-5) we need boundary conditions. In this work we are interested in the motion of the surface and, correspondingly, in free moving surface boundary conditions (for other examples, see Ref. [30]). We parameterize the boundary of the fluid as $y = h(x, t)$ assuming that the fluid domain is defined by $y \leq h(x, t)$ (see Fig. 1). The first boundary condition is

---

[1]The vorticity contribution to the pressure term of the stress tensor can fully or partially cancel the effects described in this work. The actual values of all coefficients in the stress tensor should either be measured experimentally or derived from the underlying microscopic physics.

the kinematic boundary condition (KBC) expressing the connection between the motion of the boundary of the fluid with the fluid velocity taken at the boundary

$$\partial_t h + v_x \partial_x h = v_y , \quad \text{at } y = h(x,t) . \tag{9}$$

In addition to KBC we have two dynamical boundary conditions (DBC) stating that the stress forces determined by (5) are continuous across the boundary. Here, we impose no-stress boundary conditions assuming that the fluid is in contact with a medium such as air or vacuum which cannot apply any forces except for maybe constant pressure. We have

$$T_{ij} n_j = 0 , \tag{10}$$

where the outward-pointing normal vector is given by $\mathbf{n} = \frac{1}{\sqrt{1+(\partial_x h)^2}}(-\partial_x h, 1)$.

Finally, in the case when the fluid domain is non-compact one should additionally impose boundary conditions at infinity. We will assume here that those boundary conditions are $\mathbf{v} \to 0$ as $y \to -\infty$.

## 3 Irrotational motion in the bulk

It follows from (8) that an initially irrotational region of the fluid ($\omega = 0$) stays irrotational until the vorticity is delivered to the region either by convection or by diffusion. Let us assume that the initial state of the fluid is irrotational. Then, as the fluid starts to move one should have vanishing vorticity for some finite time. Assuming that the motion of the fluid in the bulk is irrotational $\omega = 0$ we can parameterize the velocity by a potential $v_i = \partial_i \phi$ which satisfies the incompressibility condition written as the Laplace equation

$$\Delta \phi = 0 . \tag{11}$$

For such a potential flow, Eq. (7) is equivalent to Bernoulli's equation [32]

$$\partial_t \phi + \frac{1}{2}(\nabla \phi)^2 = -\tilde{p} - g y . \tag{12}$$

One can think of (12) as the equation determining the modified pressure

$$\tilde{p} = p - v_o \omega, \tag{13}$$

so that any solution of the Laplace equation depending on time as a parameter gives a local solution of the Navier-Stokes equation (7) for an incompressible flow. To fully specify the hydrodynamic flow one needs in addition to satisfy boundary conditions. Before discussing boundary conditions we notice that a potential $\phi(x,y,t)$ is harmonic inside the fluid domain $D$ and, therefore, is fully defined by its values $\phi|_\Sigma$ on a boundary of the domain $\Sigma = \partial D$. Therefore, knowing the potential at $\Sigma$ one immediately knows the potential and velocity fields everywhere inside the domain $D$. In particular, the tangent and normal components of the velocity at the boundary of the domain $\Sigma$ can be determined from one scalar function $\phi|_\Sigma$ and are not independent from each other.

In the absence of viscosities, DBC (10) reduces to a single requirement $p|_\Sigma = 0$ which can be satisfied by the proper choice of $\phi|_\Sigma$. On the other hand, when one or both viscosity coefficients $v_{e,o}$ are non-vanishing, the free surface DBC (10) produce two non-trivial boundary conditions. These conditions are impossible to satisfy just by fine tuning the scalar potential $\phi\Big|_\Sigma$.

A resolution of this problem is well known in the absence of odd viscosity but with non-zero shear viscosity $\nu_e$. The irrotationality assumption on the fluid flow should necessarily break down at the moving free surface. A time-dependent boundary layer with non-vanishing vorticity is formed at the free surface of the moving fluid [34, 43]. In this paper, we study the effects of the boundary layer on the motion of the surface in the presence of odd viscosity. Before going into this, let us first study in some detail the solution of the linearized problem applicable in the case of small amplitude surface waves.

## 4 Linear odd surface waves

Modifying the derivation of Lamb [34] for the case of non-vanishing odd viscosity, we derive a solution corresponding to surface waves in the presence of both $\nu_e$ and $\nu_o$ in the limit of small amplitude oscillations. We start by linearizing the basic equations (7) assuming that velocity, its gradients as well as the gradients of external potential are small. We obtain the following linear bulk equations:

$$\nabla \cdot \mathbf{v} = 0, \tag{14}$$
$$\partial_t \mathbf{v} = -\nabla \tilde{p} + \nu_e \Delta \mathbf{v} - g\hat{\mathbf{y}}. \tag{15}$$

These equations are identical to the equations for the fluid with $\nu_o = 0$ (with the only change $p \rightarrow \tilde{p} = p - \nu_o \omega$ and we can borrow the Lamb's solution for plane waves written using complex valued velocities (see [34], art. 349)

$$v_x = \left(A|k|e^{|k|y} + Bme^{my}\right)e^{ikx - i\Omega t}, \tag{16}$$
$$v_y = -ik\left(Ae^{|k|y} + Be^{my}\right)e^{ikx - i\Omega t}, \tag{17}$$
$$\omega = e^{ikx - i\Omega t}\left(\frac{i\Omega}{\nu_e}Be^{my}\right), \tag{18}$$
$$\tilde{p} = \Omega \frac{k}{|k|}Ae^{|k|y}e^{ikx - i\Omega t} - gy. \tag{19}$$

Here we use the convention that the actual solution is given by the real parts of these formulas. Here $A, B$ are arbitrary (complex) amplitudes and the following relation between $m$ and $k$ should be satisfied:

$$m^2 = k^2 - \frac{i\Omega}{\nu_e}. \tag{20}$$

Here we assume that $\mathrm{Re}(m) > 0$ so that the velocity vanishes in the limit $y \rightarrow -\infty$. We also linearize the DBC (10) taking the normal vector $\mathbf{n} \approx (0, 1)$

$$p = \nu_e(\partial_y v_y - \partial_x v_x) - \nu_o(\partial_x v_y + \partial_y v_x), \tag{21}$$
$$\nu_e(\partial_x v_y + \partial_y v_x) + \nu_o(\partial_y v_y - \partial_x v_x) = 0, \tag{22}$$

and also the KBC (9)

$$v_y = \partial_t h. \tag{23}$$

Substituting (17) into (23) we obtain

$$h(x, t) = \frac{k}{\Omega}(A + B)e^{ikx - i\Omega t}. \tag{24}$$

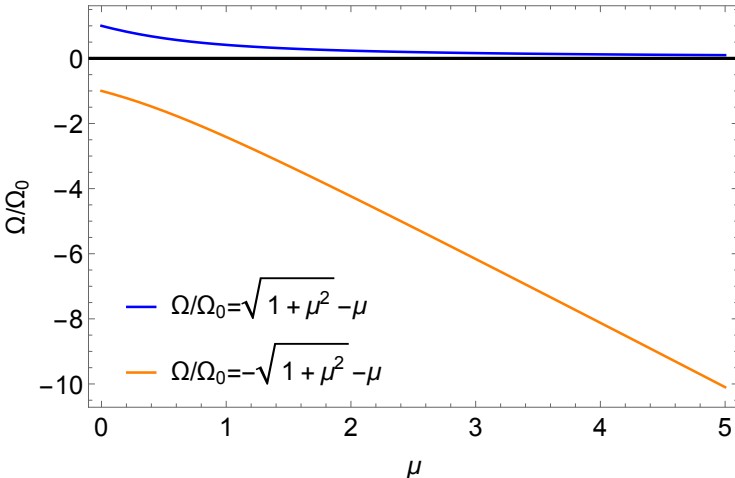

Figure 2: The dispersion of surface waves as a function of odd viscosity at vanishing shear viscosity $\nu_e \to 0$. Here $\Omega_0 = \sqrt{gk}$ and $\mu = \nu_o k^2/\Omega_0$ is a dimensionless odd viscosity.

We note here that within the linear approximation one can evaluate fields at $y = 0$ instead of $y = h(x,t)$. The only exception is the last term of (19) which should be replaced by $-gh$ at the boundary. Substituting (16,17,19,24) into the DBC (21,22) we obtain two equations for the amplitudes

$$A\Big[g|k| - \Omega^2 - 2\Omega(\nu_o k|k| + i\,\nu_e k^2)\Big] + B\Big[g|k| - 2\Omega(\nu_o k|k| + i\,\nu_e|k|m)\Big] = 0, \quad (25)$$

$$A\Big[2(\nu_o k|k| + i\,\nu_e k^2)\Big] + B\Big[\Omega + 2\nu_o km + 2i\,\nu_e k^2\Big] = 0. \quad (26)$$

Equations (25,26) together with the relation (20) fully define the solution for surface waves. In particular, the dispersion relation $\Omega(k)$ is obtained by requiring consistency of (25) and (26). The full expression for $\Omega(k)$ is not very illuminating. We analyze the dispersion relation and its different limits in Appendix A.

## 4.1 Gravity waves in the presence of small odd viscosity

In a limit of small odd and even viscosities we obtain a perturbative correction to Lamb's solution (see Appendix A)

$$\Omega \approx \pm\sqrt{g|k|} - 2i\,\nu_e k^2 - 2\nu_o k|k|. \quad (27)$$

In the limit $\nu_o = 0$ the dispersion (27) reduces to the well-known result for incompressible waves with shear viscosity [34].

It is clear from the relation (20) that there is a typical length scale $\delta = \sqrt{\nu_e/|\Omega|}$. As the frequency $\Omega$ of the surface waves remains finite in the limit $\nu_e \to 0$ we find that the $B$-component of the solution (16,17) and vorticity of the fluid (18), in particular, is localized near the surface of the fluid within the dynamic boundary layer of thickness $\delta$. For conventional slightly damped gravity waves (without odd viscosity) $\Omega \approx \pm\sqrt{g|k|}$ and the existence and structure of such a layer is well known [34,43]. In the following we will see how this boundary layer is modified by odd viscosity.

## 4.2 Surface waves dominated by odd viscosity in the absence of gravity

In the following we consider the novel regime when surface waves are dominated by the odd viscosity and the gravity is absent. Namely, we assume that $\nu_o \gg \nu_e$ and switch off the

external potential $g = 0$. The equation for the wave dispersion in the absence of gravity (see Appendix A) is:

$$\Omega^3 + 2\Omega^2 k\left(\nu_o(m + |k|) + 2i\,\nu_e k\right) + 4\Omega k^2 |k|(m - |k|)(\nu_o^2 + \nu_e^2) = 0\,. \tag{28}$$

There are only two solutions of this equation that correspond to the wave decaying into the bulk of the fluid (i.e., $\text{Re}(m) > 0$, see Fig. 2). The first one is trivial, corresponding to the arbitrary deformation of the boundary of the fluid $y = h(x)$ which is at rest: $\Omega = 0$, $\mathbf{v} = 0$. In the absence of gravity, there is no restoring force and the fluid remains motionless (we neglect surface tension in this work). However, in the presence of odd viscosity there exists also a non-trivial solution

$$\Omega \approx -2\nu_o k|k| - i\sqrt{\nu_e|\nu_o|}k^2\,, \tag{29}$$

where we keep only the leading terms in real and imaginary parts of $\Omega(k)$. The first term of (29) corresponds to a chiral wave propagating with group velocity

$$\nu_k = \frac{\partial \Omega}{\partial k} \approx -4\nu_o|k|\,. \tag{30}$$

The direction of the propagation of the wave is determined by the sign of the odd viscosity $\nu_o$. The second term (29) describes the damping of the wave. The details of the derivation of the linear wave solution and its dispersion are given in Appendix A. The limit of $\nu_e \to 0$ in (20,25,26) produces in the leading non-vanishing order

$$m \approx (1 + i\,\text{sign}(\nu_o k))\sqrt{\frac{|\nu_o|}{\nu_e}}|k|\,, \tag{31}$$

$$A \approx -2\nu_o|k|D\,, \tag{32}$$

$$B \approx -A\frac{1 - i\,\text{sign}(\nu_o k)}{2}\sqrt{\frac{\nu_e}{|\nu_o|}}\,. \tag{33}$$

The height profile and the vorticity is given by,

$$h \approx De^{ikx - i\Omega t}\,, \tag{34}$$

$$\omega = -2\nu_o k^2|k|D(1 + i\,\text{sign}(\nu_o k))\sqrt{\frac{|\nu_o|}{\nu_e}}e^{my + ikx - i\Omega t}\,. \tag{35}$$

and in the regime $\nu_e \ll \nu_o$ the vorticity is non-vanishing only in the narrow layer of the thickness of the order of $\delta = k^{-1}\sqrt{\frac{\nu_e}{|\nu_o|}}$ which can be found as $m^{-1}$ from (31). We emphasize that the diverging of vorticity as $\omega \sim 1/\sqrt{\nu_e}$ in the boundary layer is solely due the the presence of odd viscosity and is dramatically different compared to the Lamb's case where $\omega \sim O(1)$ as $\nu_e \to 0$. In the limit of interest $B \ll A$ (33) and the vertical component of the velocity is essentially defined by the $A$-component of the solution

$$v_y \approx -ikAe^{|k|y}e^{ikx - i\Omega t}\,. \tag{36}$$

On the contrary the horizontal component of the velocity has contributions from both $A$ and $B$ parts of the solution. We obtain

$$v_x \approx A|k|\left(e^{|k|y} - e^{my}\right)e^{ikx - i\Omega t}\,. \tag{37}$$

At the surface $y \approx 0$ the horizontal velocity is $\sim \sqrt{\nu_e}$ and vanishes within our accuracy (37). However, it changes rapidly with $y$ and becomes of the order of $A|k|$ at the depth of few $\delta$. This is the $v_x$ which contributes the most to vorticity so that near the surface the vorticity diverges $\omega \approx -\partial_y v_x \sim \nu_e^{-1/2}$ while the $v_x$ changes by a finite amount ($\sim \nu_e^0$) across the layer of thickness $\delta \sim \nu_e^{1/2}$. This behavior is rather different from Lamb's solution, in which only the vortical part ($B$-part) of $v_x$ vanishes as $\nu_e \to 0$.

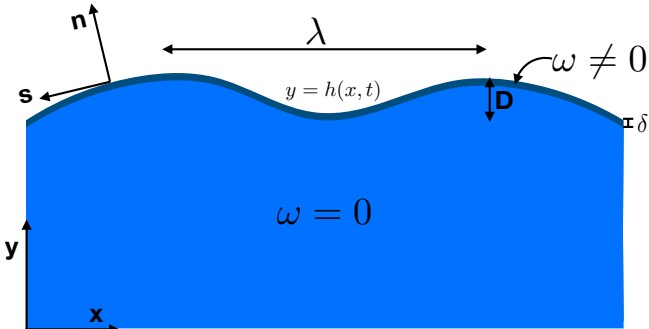

Figure 3: Surface waves with boundary layer.

## 5 Effective nonlinear boundary dynamics

In this section we present physical arguments leading to the effective nonlinear equation for surface waves relegating more precise arguments and detailed derivation to Appendix C. Let us consider a nonlinear wave in the limit of small $\nu_e$ with the amplitude $D$ small compared to the wavelength of the wave $\lambda$ but non-vanishing (weakly nonlinear waves). Here we also assume that the external gravity field is absent. In Section 4 we saw that in this limit the typical frequency $\Omega \sim \nu_o k^2$ remains finite and that the vorticity is confined to the narrow layer near the surface. Motivated by the structure of the linear solution we make the following assumptions for weakly nonlinear waves:

1. A typical time scale (frequency) remains finite in the limit of vanishing shear viscosity $\nu_e$.

2. The motion is potential (irrotational) in the bulk with vorticity localized in the narrow region near the surface — the boundary layer (see Fig. 3). The thickness of this region is much smaller than the amplitude of the wave $D$.

The main idea of the following derivation is to consider the effect of the narrow boundary layer on the potential flow of the fluid outside of the boundary layer. We essentially "integrate out" the layer and consider the potential flow with modified boundary conditions for the rest of the fluid.[2] First, let us consider the boundary layer.

Neglecting the curvature of the surface, we see from the tangent stress boundary condition (22) that in the limit $\nu_o \gg \nu_e$ we essentially have $\partial_y v_y - \partial_x v_x \approx 0$ (or $\partial_x v_x \approx 0$ using incompressibility). This means that for small curvature $D/\lambda \ll 1$ the tangent component of the fluid velocity should vanish at the free surface[3]. The odd viscosity imposes an effective no-slip boundary condition for the tangential velocity on the moving surface. The finite curvature generates a tangent velocity at the boundary but it remains small and does not invalidate the boundary layer approximation (see Appendix C). Having small tangent velocity at the boundary and finite velocity right below the boundary layer means that large vorticity is generated inside the layer (c.f., Section 4.2). The small but finite shear viscosity $\nu_e$ is important in determining the thickness of this layer but as we will see drops out of the effective equation for velocity potential. In the limit $\nu_e \to 0$ the boundary layer is infinitesimally thin and we can write the KBC (9) as

$$h_t = v_y - h_x v_x = \partial_y \phi - h_x \partial_x \phi \, . \tag{38}$$

[2]In contrast to this approach in the more precise approach of Appendix C we decompose velocity into potential and vortical part and integrate out the vortical part, again obtaining the effective equations for potential flow.

[3]More precisely, the tangent velocity at the boundary is of the second order in amplitude. For detailed scaling analysis, see Appendix. C

Here the velocity $v$ is taken at the free surface and the potential $\phi$ is taken right below the boundary layer. As $v_x$ is already of the second order of smallness in the amplitude, we can drop the term $h_x v_x$ in quadratic approximation and obtain the relation between velocity at the free surface and potential as

$$v_y \approx \partial_y \phi - h_x \partial_x \phi. \tag{39}$$

Let us now consider the normal stress boundary condition. In the limit of $\nu_e \to 0$ we can write the stress tensor (5) as

$$T_{ij} = -\tilde{p}\delta_{ij} + 2\nu_o \partial_i^* v_j. \tag{40}$$

The normal stress boundary condition at the free surface becomes

$$0 = T_{nn} = -\tilde{p} + 2\nu_o n_i n_j \partial_i^* v_j \approx -\tilde{p} - 2\nu_o(\partial_x v_y - 2h_x \partial_x v_x), \tag{41}$$

where we used $\hat{n} \approx (-h_x, 1)$ and kept only terms up to quadratic ones in $h$ and $v$. As the tangent velocity at the boundary is already of the second order in the amplitude, the term $h_x \partial_x v_x$ can be neglected in quadratic approximation. Using (39) we obtain

$$\tilde{p} = -2\nu_o \partial_x v_y = -2\nu_o \partial_x (\partial_y \phi - h_x \partial_x \phi) \tag{42}$$

as an effective dynamical boundary condition. The modified pressure does not change across the very thin boundary layer and we can use (42) as the boundary condition just below the boundary layer.

Now consider the Bernoulli equation (12) which is expected to be valid right below the boundary layer. Substituting the expression of modified pressure (42) and with $g = 0$ we have:

$$\partial_t \phi + \frac{1}{2}(\nabla \phi)^2 = 2\nu_o \partial_x (\partial_y \phi - h_x \partial_x \phi). \tag{43}$$

This equation is assumed to be valid at the surface $y = h(x, t)$, where we neglect the thickness of boundary layer and essentially replace the true physical surface of the fluid by the inner surface of the boundary layer. To proceed we introduce the boundary value of the potential

$$\tilde{\phi}(x, t) = \phi(x, h(x, t), t).$$

We find in quadratic approximation

$$\partial_t \tilde{\phi} = \partial_t \phi + h_t \partial_y \phi \approx \partial_t \phi + (\partial_y \phi)^2,$$

and substituting into (43)

$$\partial_t \tilde{\phi} + \frac{1}{2}(\partial_x \phi)^2 - \frac{1}{2}(\partial_y \phi)^2 = 2\nu_o \partial_x (\partial_y \phi - h_x \partial_x \phi). \tag{44}$$

Finally, we remember that the potential $\phi$ is harmonic in the lower half plane (strictly speaking, it is harmonic for $y \leq h(x, t)$). Rewriting the above equation in terms of $\tilde{\phi}(x, t)$ (see Eq. 102 and Appendix B for details) and keeping up to quadratic terms we obtain,

$$\tilde{\phi}_t + \frac{1}{2}(\tilde{\phi}_x^2 - (\tilde{\phi}_x^H)^2) + 2\nu_o \tilde{\phi}_{xx}^H = -2\nu_o \left[ h\tilde{\phi}_x + (h\tilde{\phi}_x^H)^H \right]_{xx}. \tag{45}$$

Here "$H$" denotes Hilbert transform [4] defined as

$$\phi^H(x) = P.V. \frac{1}{\pi} \int_{-\infty}^{+\infty} \frac{\phi(x')dx'}{x' - x}. \tag{46}$$

---

[4] If the function $\phi(x, y)$ is harmonic in the lower half plane $y \leq 0$, the complex function $2\phi^- \equiv \phi + i\phi^H$ is analytic in the lower half plane of $z = x + iy$. Its real and imaginary parts satisfy the Cauchy-Riemann relations $\partial_x \phi = \partial_y \phi^H$ and $\partial_y \phi = -\partial_x \phi^H$. The relations used in the text are derived in Appendix B. Appendix includes this derivation to next order correction in nonlinearity. These corrections are due to the difference between $\phi(x, 0)$ and $\phi(x, h(x, t)) \equiv \tilde{\phi}(x, t)$ and between the true domain of harmonicity $y \leq h(x)$ and the lower half plane $y \leq 0$.

The KBC (38) becomes

$$h_t + \tilde{\phi}_x^H = -\left[h\tilde{\phi}_x + (h\tilde{\phi}_x^H)^H\right]_x. \tag{47}$$

This form does not depend on $\nu_o$ and coincides with Kuznetsov et al. [42]. In terms of $\tilde{u} = \tilde{\phi}_x$ it becomes

$$h_t + \tilde{u}^H + \left[h\tilde{u} + (h\tilde{u}^H)^H\right]_x \approx 0. \tag{48}$$

Differentiating (45) with respect to $x$ and introducing $\tilde{u} = \tilde{\phi}_x$ we obtain from (45,47) the following system of equations:

$$\tilde{u}_t + \tilde{u}\tilde{u}_x - \tilde{u}^H\tilde{u}_x^H + 2\nu_o\tilde{u}_{xx}^H = -2\nu_o\left[h\tilde{u} + (h\tilde{u}^H)^H\right]_{xxx}, \tag{49}$$

$$h_t + \tilde{u}^H = -\left[h\tilde{u} + (h\tilde{u}^H)^H\right]_x. \tag{50}$$

This nonlinear system is Hamiltonian! It is remarkable that the Hamiltonian itself is the same as for the fluid without odd viscosity and is equal to the total kinetic energy of the fluid. However, the Poisson brackets of the fields are modified by odd viscosity dependent terms. For details see next section.

In the small surface angle approximation (see Ref. [42]) we neglect the right hand side of (49) and (50) and obtain

$$\tilde{u}_t + \tilde{u}\tilde{u}_x - \tilde{u}^H\tilde{u}_x^H + 2\nu_o\tilde{u}_{xx}^H = 0, \tag{51}$$

$$h_t + \tilde{u}^H = 0. \tag{52}$$

The first equation reminds us of the well-known Benjamin-Davis-Ono (BDO)[5] equation but the nonlinear term is very different. This equation results in a new class of non-linear chiral dynamics which we dub as chiral Burgers equation (complex Burgers equation with an additional analyticity condition, see section 7). We notice here that linearizing (49,50) and including gravity we obtain

$$\tilde{u}_t + 2\nu_o\tilde{u}_{xx}^H + gh_x = 0, \tag{53}$$

$$h_t + \tilde{u}^H = 0. \tag{54}$$

It is easy to check that these linear equations produce the dispersion given by (93). Also, in the absence of gravity $\tilde{u}_t = 2\nu_o(h_{xx})_t$ so that the tangent velocity at the bottom of the boundary layer changes in proportion to the curvature of the surface $-h_{xx}$.

To summarize, the tangent stress boundary condition is satisfied through the formation of the vortical boundary layer at the surface of the fluid. Right below the boundary layer the modified pressure is given by $\tilde{p} \approx -2\nu_o\partial_x\nu_y$. Using the potentiality of the flow in the bulk below the boundary layer we arrived at the Hamiltonian system (49,50). An additional small surface angle approximation gives a simplified system[6] (51,52).

## 6 Hamiltonian structure

As described in the previous section, in the limit $\nu_e \to 0$ the dissipation vanishes and we expect the effective surface dynamics equations to have a Hamiltonian structure. In this section, we

---

[5]The Benjamin-Davis-Ono equation has a form $u_t + uu_x + u_{xx}^H = 0$.

[6]Unfortunately, the simplified system (51,52) is not Hamiltonian.

show that this is indeed so. Let us first consider the total kinetic energy of the fluid[7]

$$E = \int dx \int_{-\infty}^{h(x)} dy \frac{1}{2}(\phi_x^2 + \phi_y^2) = \int dx \int_{-\infty}^{h(x)} dy \frac{1}{2}\Big((\phi\phi_x)_x + (\phi\phi_y)_y\Big).$$

Here we used the incompressibility condition in the bulk $\phi_{xx} + \phi_{yy} = 0$. We proceed by applying Green's theorem and using (102)

$$E = \int dx \left\{ -\frac{1}{2}h_x \phi \phi_x + \frac{1}{2}\phi\phi_y \right\}_{y=h} = \int dx \frac{1}{2}\tilde{\phi}(\partial_y \phi - h_x \partial_x \phi)_{y=h}$$

$$\approx \int dx \frac{1}{2}\tilde{\phi}\Big( -\tilde{\phi}_x^H - (h\tilde{\phi}_x)_x - (h\tilde{\phi}_x^H)_x^H \Big) = \int dx \left\{ \frac{1}{2}h\big[(\tilde{\phi}_x)^2 - (\tilde{\phi}_x^H)^2\big] - \frac{1}{2}\tilde{\phi}\tilde{\phi}_x^H \right\}. \quad (55)$$

The obtained expression coincides with the one of Ref. [42]. Let us now consider a Hamiltonian which is given by the expression (55) with the addition of gravitational potential energy

$$H = \int dx \left\{ h\Big[\frac{1}{2}(\tilde{\phi}_x)^2 - \frac{1}{2}(\tilde{\phi}_x^H)^2\Big] - \frac{1}{2}\tilde{\phi}\tilde{\phi}_x^H + \frac{1}{2}gh^2 \right\}. \quad (56)$$

We can easily calculate the variation of the Hamiltonian (56) to obtain

$$\delta H = \int dx \left\{ \delta h\Big[\frac{1}{2}(\tilde{\phi}_x)^2 - \frac{1}{2}(\tilde{\phi}_x^H)^2 + gh\Big] - \delta\tilde{\phi}\Big[\tilde{\phi}^H + h\tilde{\phi}_x + (h\tilde{\phi}_x^H)^H\Big]_x \right\}. \quad (57)$$

We also consider slightly non-canonical Poisson brackets

$$\{\tilde{\phi}, h'\} = \delta(x - x'), \quad (58)$$

$$\{\tilde{\phi}, \tilde{\phi}'\} = -2\nu_o \partial_x \delta(x - x'), \quad (59)$$

$$\{h, h'\} = 0. \quad (60)$$

Instead of verifying Jacobi identity for brackets (58-60) we notice that defining

$$\Phi = \tilde{\phi} - \nu_o h_x \quad (61)$$

we obtain canonical Poisson brackets between $h$ and $\Phi$.

The equations of motion generated by the Hamiltonian $H$ with the use of these Poisson brackets are

$$\tilde{\phi}_t + \frac{\delta H}{\delta h} = 2\nu_o \partial_x \frac{\delta H}{\delta\tilde{\phi}}, \quad h_t = \frac{\delta H}{\delta\tilde{\phi}}. \quad (62)$$

The equations of motion (62) from the above variational formula (57) reproduces (49,50) with $g$. Using (50) we can write the first equation (49) in a more compact form

$$\tilde{\phi}_t + \frac{1}{2}(\tilde{\phi}_x^2 - (\tilde{\phi}_x^H)^2) + gh = 2\nu_o h_{xt}. \quad (63)$$

## 6.1 Conservation laws

In this section, we demonstrated the existence of a Hamiltonian structure to the non-linear dynamics up to the quadratic order. This automatically implies the integral of motion of (49,50)

---

[7]We neglect the kinetic energy of the vortical layer as it is vanishing in the limit $\nu_e \to 0$.

given by (56). In addition to energy, it is easy to check that the following quantities are conserved by (49,50)

$$M = \int_{-\infty}^{\infty} dx\, h, \tag{64}$$

$$P = \int_{-\infty}^{\infty} dx\, h(\tilde{\phi}_x - v_0 h_{xx}) = \int_{-\infty}^{\infty} dx\, h\Phi_x. \tag{65}$$

The first one is easily recognized as a conservation of fluid mass. The conservation law (65) is a consequence of translational invariance of equations (49,50). Indeed, remembering the canonical Poisson brackets of $h$ and $\Phi$ in Eq. 61 one can easily recognize in (65) the generator of translations in the $x$ direction. The quantity $P$ appearing in the effective one-dimensional equations is related to the pseudomomentum of the two-dimensional fluid. The pseudomomentum conservation arises when the wave disturbance is translated without translating the medium in which it exists. This is a symmetry only when both the background space and the medium are homogeneous. [44–46].

## 7 Nonlinear surface waves: Chiral Burgers equation

We now show that the small angle dynamics of the parent Hamiltonian system (51) results in what we call chiral Burgers equation. We introduce $\tilde{u} = u(x) + \bar{u}(x)$, where $u(z)$ is the function analytic in the lower half-plane of $z$ and $\bar{u}$ its complex conjugate on real axis. Then the Hilbert transform (46) is equivalent to $\tilde{u}^H = i(\bar{u} - u)$. The equation (51) can be decoupled into the equations for the parts holomorphic in the lower and upper half-planes. In particular for the part holomorphic in the lower half-plane we obtain (2) which we reproduce here for the reader's convenience

$$u_t + 2uu_x - 2i v_o u_{xx} = 0. \tag{66}$$

This is a complex Burgers equation with imaginary viscosity, i.e. coefficient in front of the Burgers term $u_{xx}$. The complex Burgers equation is a simple example of a nonlinear dispersive equation. It was described in [47] with no physical connections. In addition to (66) we have an additional analytic constraint on the solution $u(x,t)$. The function $u(x,t)$ is subject to an additional condition that it is equal on the real axis to the function $u(z,t)$ analytic in the lower half-plane. This analyticity requirement enforces chiral dynamics and hence results in chiral Burgers equation. We discuss this equation in more detail in Appendix D. Here we just give a few examples of the solutions of (66). Let us start with the periodic nonlinear wave solution

$$u = \frac{U}{1 - e^{ik(x - Ut) + ka}}, \tag{67}$$

where the velocity of the wave is given by

$$U = -2v_o k \tag{68}$$

and we restrict $k > 0$ which guarantees the analyticity of (67) in the lower half of complex plane. The solution is characterized by two parameters: the wave vector $k > 0$ and the parameter $a > 0$.

The real field $\tilde{u} = u + \bar{u}$ is then given by,

$$\tilde{u} = U\left(1 - \frac{\sinh(ka)}{\cosh(ka) - \cos(k(x - Ut))}\right). \tag{69}$$

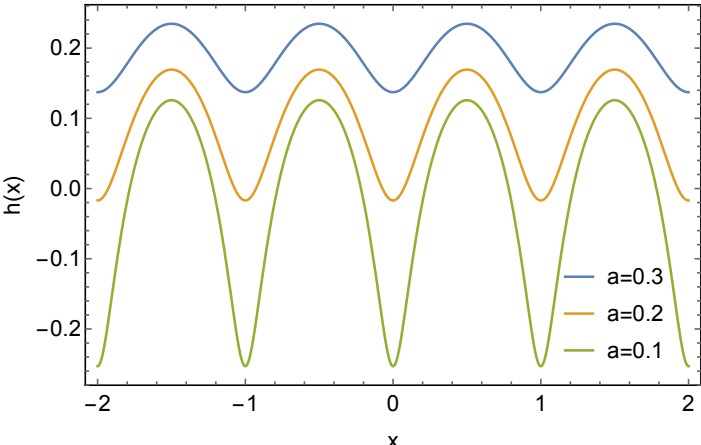

Figure 4: Nonlinear surface wave profiles $h(x)$ given by Eq. 70 for $k = 2\pi$ and $a = 0.1, 0.2, 0.3$ from bottom to top.

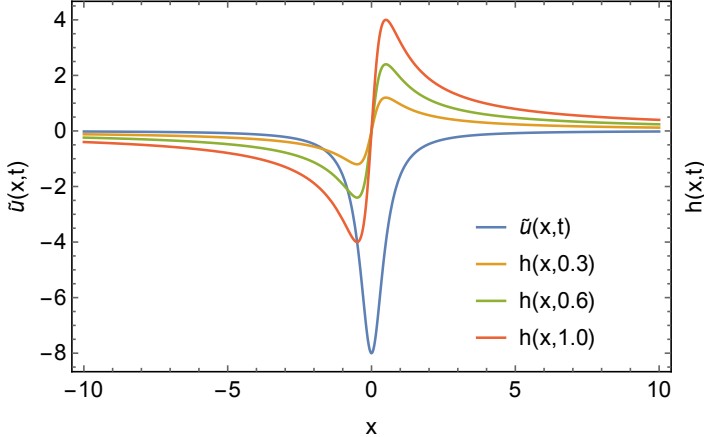

Figure 5: Nonlinear surface wave profiles $\tilde{u}(x), h(x,t)$ for different times $t = 0.3, 0.6, 1.0$.

We can also find the profile of the wave $h(x,t)$ from $\partial_t h = \partial_x \tilde{\phi} = i(u - \bar{u})$. We obtain integrating (67) over time (up to an additive constant)

$$h = \frac{1}{k} \ln \left[ \cosh(ka) - \cos(k(x - Ut)) \right]. \tag{70}$$

In the limit of $a \to \infty$ while keeping $k > 0$ fixed, we obtain,

$$\tilde{u} \approx D\Omega \cos(kx - \Omega t), \quad h \approx D \cos(kx - \Omega t). \tag{71}$$

Here the amplitude of the wave $D = -2k^{-1}e^{-ka}$ and $\Omega = -2v_o k^2$. In this limit we recover the linearized Lamb solution of Section 4.2. Fixing $a$ and taking a limit $k \to 0$ we obtain the time independent solution for $u$ as

$$u = -\frac{2i v_o}{x + ia}, \quad \tilde{u} = -\frac{4 v_o a}{x^2 + a^2}. \tag{72}$$

This solution is reminiscent of the well known single-soliton solutions of the BDO equation[7]. However, in this limit we obtain the following expression for the height profile (see Fig. 5)

$$h = \frac{4 v_o x}{x^2 + a^2} t. \tag{73}$$

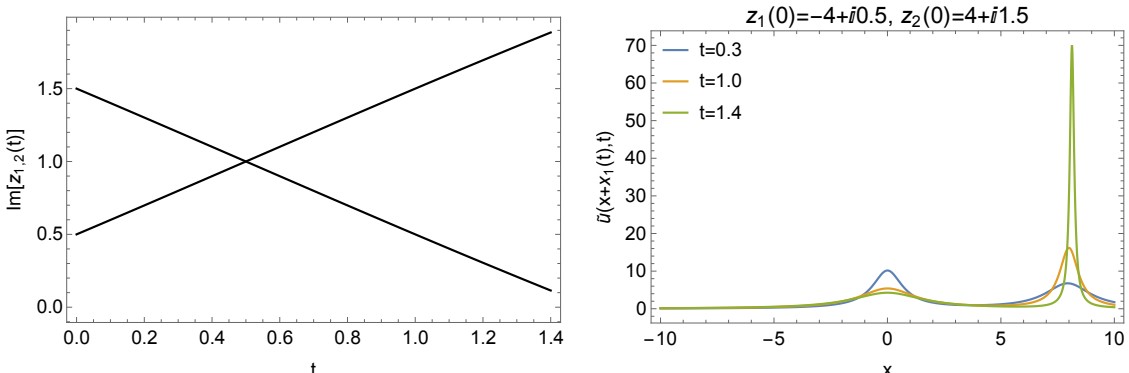

Figure 6: (Left) Dynamics of the imaginary part of the two poles given by Eq. 75. One of the poles hits the real axis at finite time. (Right) Two soliton profile $\tilde{u}(x,t)$ for different times $t = 0.3, 1.0, 1.4$ governed by the pole dynamics plotted on the left. One of the soliton profiles widens whose pole moves away from the real axis and the other soliton keeps sharpening while moving to the right (relative to the first soliton) and diverges at the time when the pole hits the real axis.

This solution grows in time linearly. In time of the order of $t_* = a^2/\nu_o$ the curvature of the profile becomes significant and the assumptions of small curvature used in deriving (66) are violated. It is known that complex Burgers equation without the analyticity requirement (66) possesses multi-pole solutions [48]. In Appendix E we present some of multi-pole solutions. Here we discuss just a two-pole solution. It is given by

$$u(x,t) = -2i\,\nu_o\left(\frac{1}{x - z_1(t)} + \frac{1}{x - z_2(t)}\right), \tag{74}$$

where

$$\dot{z}_{1,2}(t) = \mp\frac{4i\,\nu_o}{z_1 - z_2}. \tag{75}$$

This solution is plotted in Figure 6 for several times. It essentially looks like two lumps which move and change their width. One of them is spreading while the other one is shrinking. At some finite time one of the complex poles hits the real axis and one cannot assume analyticity of $u(x)$ in the lower half complex plane beyond that time.

## 8   Discussion and Outlook

In this work we considered the motion of a free surface of an incompressible fluid with odd viscosity. For such an incompressible fluid the effect of the odd viscosity reduces to a modification of boundary conditions at the surface.

We solved the full system of hydrodynamic equations with proper boundary conditions in the linear approximation in Section 4. This solution generalizes the well known Lamb's result for linear surface gravity waves with ordinary shear viscosity [34]. Similar to the Lamb's solution, the motion of the fluid is mostly potential in the bulk with a narrow vorticity layer formed at the surface due to the tangent stress present in viscous fluids. The structure of the boundary layer is, however, different due to the presence of odd viscosity. The width of the boundary layer is $\sim \sqrt{\nu_e}$ as in Lamb's case, but vorticity in the layer diverges as $1/\sqrt{\nu_e}$ in the limit of vanishing shear viscosity $\nu_e \to 0$. The latter is in contrast with conventional Lamb's

result where vorticity remains finite even within the boundary layer. Unlike the shear viscosity, the odd viscosity is non-dissipative and results in a real correction to the dispersion of surface waves. We computed these corrections in different limits. Remarkably, at finite odd viscosity the surface waves can propagate even in the absence of gravity with the dispersion of linear waves given by Eq. 1. The corresponding wave is chiral with the direction of the propagation defined by the sign of the odd viscosity coefficient $\nu_o$. In this paper we focused on the case when gravity is absent.

The motion of the fluid corresponding to surface waves is potential except for the narrow boundary layer at the surface. One might try to represent the effect of such a boundary layer as modified boundary conditions imposed on irrotational (potential) fluid. We derived such boundary conditions and obtained a system of nonlinear Hamiltonian equations governing the dynamics of free surface. Remarkably, in the limit of vanishing shear viscosity, the only effect of the odd viscosity is a shift in the Poisson brackets (Eqs. 58 59 and 60) without any changes to the Hamiltonian (compare to Ref. [27]). In total we obtain three conservation laws corresponding to energy, mass and (pseudo)momentum. The dynamics corresponding to the Hamiltonian within small surface angle approximation reduces to the chiral Burgers equation with imaginary viscosity (2) as the effective one-dimensional equation governing weakly nonlinear surface waves in Section 5 and Appendix C. The chiral Burgers term results in non-singular solution in the form of periodic non-linear waves in contrast to the inviscid limit where almost all initial condition results in finite time singularities [42]. However, even for chiral Burgers case generic pole solutions result in finite time singularities (see Appendix E). It would be interesting to study if and how the full non-linear Hamiltonian system alters these small angle finite time singularities. We leave this interesting study for future.

The complex Burgers equation has attracted some attention from a purely mathematical point of view as a simple example of a nonlinear dispersive equation [47,48]. However, to the best of our knowledge the chiral Burgers dynamics of surface waves caused by odd viscosity is the first physical application of such an equation.

For vanishing shear viscosity, the vortical boundary layer becomes infinitesimally thin. With divergent vorticity, it is essentially equivalent to the discontinuity of the tangent velocity at the boundary. This means that the solution of the hydrodynamic equations for $\nu_e = 0$ is a weak solution with a discontinuity of the tangent component of fluid velocity at the boundary. The vorticity is generated through odd viscosity by the motion of the boundary of the fluid.[8] When shear viscosity is finite but very small, its only effect is to spread the vorticity layer to make its width $\sim \sqrt{\nu_e}$. In fact, instead of considering finite shear viscosity to regularize the weak solution one might put it to zero and instead consider the fluid with small but finite compressibility. In this case, the incompressible limit corresponds to the limit of infinite sound velocity $\nu_s \to \infty$ and the width of the boundary layer scales roughly as $\sim 1/\nu_s$. However, the structure of the boundary layer is different in this case. While the dispersion of linear surface waves in the limit $\nu_s \to \infty$ remains (1), finding the effective nonlinear equation is still an open question. We present an analysis of weakly compressible case elsewhere.

It is known that the complex Burgers equation (2) possesses multi-pole solutions [48]. Not all of these solutions are acceptable for the chiral Burgers dynamics of surface waves considered here. In fact, the chiral Burgers equation with additional analytic requirements comes from the real integro-differential equation (51). This equation belongs to a one-parameter family of integro-differential equations together with the Benjamin-Davis-Ono equation (see Appendix F). It deserves to be studied separately in more detail.

We consider some of the exact solutions of (2) appropriate to our boundary conditions

---

[8]This is similar to the known fact that in compressible fluids the vorticity is generated by compression through the odd viscosity [26]. In the incompressible case, the motion of the boundary plays the role of compression generating vorticity.

in Section 7 and Appendices D,E. We find that some solutions, such as the periodic solution (67,69,70) are well behaved. On the other hand the multipole solutions exist only for finite time of the order $t_* = a^2/\nu_0$, where $a$ is some typical length scale. Within this time, one of the complex poles characterizing the solution hits the real axis, which corresponds to one of the bumps becoming very nonlinear. Close to this time the main assumption of weak nonlinear corrections we made in this work are violated. We cannot trust the chiral Burgers equation about and beyond this point in time. The behavior of the surface wave at longer times remains an open problem. There are a few scenarios that seem to be possible. One is that the equation can be corrected by terms higher order in derivatives and nonlinearity and those corrections prevent solutions from getting sharp and highly nonlinear. Another is that beyond $t_*$ the assumption of the vorticity confined to a boundary layer breaks severely and the layer is destroyed. Then the full two-dimensional dynamics should be used. The final scenario seems to be the most attractive to us. In this scenario the vorticity stops to be confined to a boundary layer but in a nice way. Namely, the imaginary pole hitting the real axis (surface of the fluid) penetrates inside the fluid forming a real vortex of quantized vorticity proportional to $\nu_o$. One can check this conjecture by repeating our derivations not assuming the full potentiality of the fluid in the two-dimensional bulk but allowing point vortices in the bulk. We postpone this study to the future.

We would like to conclude with a remark that while the study of surface waves in incompressible fluid with odd viscosity was partially motivated by hydrodynamics of quantum Hall states, the hydrodynamic equations considered here are not directly relevant to quantum Hall physics. While quantum Hall states are thermodynamically incompressible, they correspond to a vanishing sound velocity limit due to the gap in the excitation spectrum (in contrast to infinite sound velocity limit considered here). Of course, in addition to this subtle difference, the external magnetic field should definitely be included into quantum Hall hydrodynamics. We believe that the hydrodynamics equations considered in [11] might serve as a good starting point to investigate surface modes of quantum Hall states. More directly, the results of this work should be applicable to chiral active liquids which are expected to have non-vanishing odd viscosity [29]. In those applications one should also consider various types of damping terms such as rotational viscosity as well as other parity violating terms.

**Acknowledgements**   We are grateful to Gustavo Monteiro, William Irvine, Paul Wiegmann, and Aleksandr Bogatskiy for many fruitful discussions related to this project. AGA's work was supported by grants NSF DMR-1606591 and US DOE DESC-0017662.

# A Dispersion relation for linear surface waves

In this appendix, we discuss in more detail the dispersion relation for surface waves in various limits. The starting point for this analysis is Eqs. (25) and (26) which combined with (20) yields the dispersion $\Omega(k)$. We note that the linear solution (16-26) has a symmetry $k \to -k, \Omega \to -\bar{\Omega}, m \to \bar{m}$ etc., where bar denotes complex conjugation. Because of this symmetry, to avoid double counting we will keep $k > 0$ in this section. Another symmetry which follows from the consistency condition is $k \to -k, v_o \to -v_o, \Omega \to \Omega, m \to m$. This indicates that we can also stick to positive $v_o = |v_o|$. Then, the consistency of (25,26) gives an equation for the dispersion

$$\frac{gk - \Omega^2 - 2\Omega k^2(v_o + i v_e)}{2k^2(v_o + i v_e)} = \frac{gk - 2\Omega k(v_o k + i v_e m)}{\Omega + 2k(v_o m + i v_e k)}, \tag{76}$$

$$m^2 = k^2 - \frac{i\Omega}{v_e}, \quad \mathrm{Re}(\tilde{m}) > 0. \tag{77}$$

The first equation can be turned into a sixth order polynomial equation for $m$ using the second equation.

One possible solution of this equation, which can be verified by direct inspection, is $m = k$ and $\Omega = 0$ – a stationary mode. We can factorize the polynomial equation for $m$ to exclude this solution. The result is a fifth order equation. Introducing the dimensionless quantities

$$\beta^2 = \frac{v_e k^2}{\sqrt{gk}}, \quad \alpha = \frac{v_o}{v_e}, \quad \kappa = m/k, \quad \Omega = i v_e k^2(\kappa^2 - 1), \tag{78}$$

we express the equation for the dispersion in the following economical form

$$\beta^{-4}(\kappa + 1 - 2i\alpha) + (\kappa + 1)\left[\kappa^2 + 1 - 2i\alpha\right]\left[\kappa^2 + 1 - 2i\alpha\kappa\right] = 4(\kappa + 1)(\kappa - i\alpha)(1 - i\alpha). \tag{79}$$

We will proceed now to solve (79) perturbatively in various limits.

## A.1 Overdamped Waves

In the overdamped regime with $g = 0$ and $v_o = 0$, the problem reduces to

$$\kappa^4 + 2\kappa^2 + 1 - 4\kappa = 0, \tag{80}$$

which actually has $\kappa = 1$ as a solution. This means the stationary zero mode is doubly degenerate. There is an additional root with real positive part $\mathrm{Re}(\kappa) > 0$. This means there is a single overdamped mode with imaginary quadratic dispersion

$$\Omega = -i v_e \left(1 - \kappa_0^2\right) k^2, \tag{81}$$

$$\kappa_0 = \frac{1}{3}\left(-1 - \frac{4 \times 2^{2/3}}{\left(13 + 3\sqrt{33}\right)^{1/3}} + 2^{1/3}\left(13 + 3\sqrt{33}\right)^{1/3}\right) \approx 0.295598. \tag{82}$$

As anticipated, these waves are damped since $\mathrm{Im}(\Omega) < 0$. This solution is perhaps remarkable for how complicated the numerical factor ends up being for the simplest case of an overdamped viscous wave. Note that $|1 - \kappa_0^2| < 1$.

## A.2 Gravity Dominated Surface Waves

Next, we consider a regime where $\beta << 1$, i.e. which is gravity dominated, and look at even and odd viscosity corrections to the classical gravity wave dispersion.

In order to fruitfully analyze the polynomial equation (79), we will rescale $\kappa = \beta^{-1} x$ and consider small $\beta$ solutions of

$$(x + \beta - 2i\alpha\beta) + (x + \beta)\left[x^2 + \beta^2 - 2i\alpha\beta^2\right]\left[x^2 + \beta^2 - 2i\alpha\beta x\right]$$
$$= 4\beta^3(x + \beta)(x - i\alpha\beta)(1 - i\alpha). \quad (83)$$

**Zero Viscosity** To recover the classical dispersion, we simply set $\beta = 0$ in the equation above, yielding $x^4 = -1$ and use $\Omega = \frac{i\nu_e k^2}{\beta^2}\left(x^2 - \beta^2\right) \rightarrow i\sqrt{gk}\, x^2$ to give dispersion for deep gravity waves

$$\Omega = \pm\sqrt{gk}, \quad (84)$$

In this limit, there is only a single vertical length scale, so $m$ completely drops out of the problem.

**Lamb's Solution** Keeping the odd viscosity zero, but turning on a small shear viscosity amounts to considering $\alpha = 0$ and small $\beta << 1$ corrections to the dispersion (84). Apparently, $x = -\beta$ is a solution. But since this is unphysical (negative solutions do not decay into the bulk), we simply factor it out and are left with

$$1 + (x^2 + \beta^2)^2 = 4\beta^3 x, \quad (85)$$

The decaying solutions have the perturbative expansion

$$x_\pm = e^{\pm i\pi/4} \pm \frac{i\beta^2}{2}e^{i\pi/4} + O(\beta^3), \quad (86)$$

which leads to the dispersion

$$\Omega_\pm = \mp\sqrt{gk} - 2i\,\nu_e k^2 + O(\nu_e^{3/2}). \quad (87)$$

Curiously, damped gravity waves decay faster than the overdamped mode (81).

**Odd Corrections to Lamb's solution** We consider corrections to Lamb's solution when odd viscosity is the smallest scale, i.e. $\alpha << \beta << 1$. In this case, there are again two propagating modes

$$x_1 = -e^{3i\pi/4} + \frac{1}{2}\beta^2 e^{3i\pi/4}(i + 2\alpha) - i\beta^3(i + \alpha)^2 + O(\beta^4), \quad (88)$$

$$x_2 = e^{i\pi/4} + \frac{1}{2}\beta^2 e^{i\pi/4}(i + 2\alpha) + i\beta^3(i + \alpha)^2 + O(\beta^4). \quad (89)$$

$$\Omega_1 = \sqrt{gk} - 2i\,\nu_e k^2 - 2\nu_o k^2 + O(\nu_e^{3/2}), \quad (90)$$
$$\Omega_2 = -\sqrt{gk} - 2i\,\nu_e k^2 - 2\nu_o k^2 + .... \quad (91)$$

This is technically a perturbation series in $\beta$ having fixed $\alpha$. Therefore, it is valid also in the regime where $\nu_e k^2 \ll \nu_o k^2 \ll \sqrt{gk}$.

**Zero Shear Viscosity** Finally, we find the dispersion in the absence of shear viscosity but without any additional assumptions on gravity and odd viscosity. We take a limit $\alpha \to \infty$, $\beta \to 0$ keeping

$$\mu = \alpha\beta^2 = \frac{\nu_o k^2}{\sqrt{gk}} = const. \tag{92}$$

We obtain from (83) the equation $x^4 - 2i\mu x^2 + 1 = 0$ and corresponding dispersion

$$\Omega = \sqrt{gk}\left(\pm\sqrt{1+\mu^2} - \mu\right). \tag{93}$$

The latter formula produces (84) for $\mu = 0$ ($\nu_o = 0$) and gives corrections to Lamb's solution (90,91) for $\nu_e = 0$. However, Eq. 93 can also be used when gravity is negligible producing in this limit $\Omega = \{0, -2\nu_o k^2\}$ (c.f., with the next section).

### A.3 Viscosity Dominated Surface Waves

The main case considered in the bulk of the paper consists of viscosity dominated waves with $\nu_e \ll |\nu_o|$, and $g = 0$. For zero gravity, the equation (79) simplifies considerably. In fact, using $\Omega = \tilde{\Omega}k^2$, we find that $\tilde{\Omega}$ is independent of $k$ and given by

$$\frac{-\tilde{\Omega}^2 - 2\tilde{\Omega}(\tilde{\nu}_o + i\nu_e)}{2(\tilde{\nu}_o + i\nu_e)} = \frac{-2\tilde{\Omega}(\tilde{\nu}_o + i\nu_e\tilde{m})}{\tilde{\Omega} + 2(\tilde{\nu}_o\tilde{m} + i\nu_e)}, \quad \tilde{m} = \sqrt{1 - i\tilde{\Omega}/\nu_e}. \tag{94}$$

We find that the leading dispersion in the limit $\nu_e \ll |\nu_o|$ is given by

$$\Omega \approx -2\nu_o k^2 - (i - \text{sign}(\nu_o))k^2\sqrt{|\nu_o|\nu_e} - \frac{3}{2}i\nu_e k^2 + O(\nu_e^{3/2})$$

$$\approx -2\nu_o k^2 - ik^2\sqrt{|\nu_o|\nu_e}, \tag{95}$$

which describes a chiral wave propagating in the direction set by the sign of $\nu_o$. The boundary layer is characterized by

$$m = k\sqrt{\frac{|\nu_o|}{\nu_e}}\left(1 + i\,\text{sign}(\nu_o) - \frac{1}{2}\sqrt{\frac{\nu_e}{|\nu_o|}} + \frac{3}{16}(i\,\text{sign}(\nu_o) - 1)\frac{\nu_e}{|\nu_o|} + O(\nu_e^{3/2})\right), \tag{96}$$

showing that the width of the boundary layer for disturbances of characteristic length $\lambda$ scales like $\lambda\sqrt{\nu_e/|\nu_o|}$. Furthermore, we see that the imaginary part will lead to a rapidly varying phase in the vertical direction.

## B Harmonic functions and Hilbert transform

A function $\phi(z = x + iy)$ harmonic for $y \leq h(x)$ can be written as

$$\phi(z) = \text{Re}\,\frac{1}{\pi}\int_{-\infty}^{+\infty} dx'\rho(x')\log\left(z - x' - ih(x') - i0\right), \tag{97}$$

where $i0$ is equivalent to $i\epsilon$ with $\epsilon \to +0$. We calculate

$$(\partial_y\phi)_{x+ih} = \text{Re}\,\frac{i}{\pi}\int_{-\infty}^{+\infty} dx'\rho(x')\frac{1}{x - x' + i(h(x) - h(x')) - i0}$$

$$\approx -\rho + h\rho_x^H - (h\rho)_x^H + O(h^2), \tag{98}$$

where we have used the definition of Hilbert transform given in Eq. 46 and we used Plemelj formula

$$\frac{1}{x-x'-i0} = P.V.\frac{1}{x-x'} + i\pi\delta(x-x'),$$

such that

$$\frac{1}{\pi}\int dx'\frac{\rho(x')}{x-x'-i0} = -\rho^H(x) + i\rho(x).$$

Then we introduce $\tilde{\phi}(x) = \phi(x,h(x))$ and compute

$$\partial_x\tilde{\phi} = \mathrm{Re}\,\frac{1}{\pi}\int_{-\infty}^{+\infty} dx'\rho(x')\frac{1+ih_x(x)}{x-x'+i(h(x)-h(x'))-i0} \approx -\rho^H + O(h^2). \tag{99}$$

Comparing these two equations we obtain to linear order in $h$

$$(\partial_y\phi)_{x+ih} = -\tilde{\phi}_x^H - h\tilde{\phi}_{xx} - (h\tilde{\phi}_x^H)_x^H. \tag{100}$$

This form coincides with Kuznetsov et al. [42]. For completeness we also give

$$(\partial_x\phi)_{x+ih} = \tilde{\phi}_x + h_x\tilde{\phi}_x^H \tag{101}$$

and another useful formula valid to linear order in $h$

$$(\partial_y\phi - h_x\partial_x\phi)_{x+ih} = -\tilde{\phi}_x^H - (h\tilde{\phi}_x)_x - (h\tilde{\phi}_x^H)_x^H. \tag{102}$$

## B.1 Useful formulas

Using the decomposition $f^H = i(f^+ - f^-)$, where $f^\pm$ is the function analytic in the upper(lower) half of complex plane one can derive the following identity:

$$(fg)^H - (f^Hg^H)^H = f^Hg + fg^H. \tag{103}$$

Another useful formula

$$f^Hg^H - (f^Hg)^H = (f^+g^-)^+ + (f^-g^+)^-. \tag{104}$$

We conclude this section by noticing that the integral of total Hilbert transform vanishes

$$\int_{-\infty}^{+\infty} f^H(x)\,dx = 0. \tag{105}$$

Applying this to (103) we can "integrate by parts" expressions with Hilbert transform

$$\int_{-\infty}^{+\infty} f(x)g^H(x)\,dx = -\int_{-\infty}^{+\infty} f^H(x)g(x)\,dx. \tag{106}$$

# C  Quasi-potential approximation for nonlinear surface waves

The goal of this section is to derive an effective nonlinear description of surface waves in the absence of gravity and shear viscosity. The main idea is to "integrate out" the vortical component of the fluid leaving equations that contain only the potential part of the flow – "quasi-potential approximation". Technically we follow the works [37, 39]. However, our results are very different from those works as the main physics is governed by the odd viscosity and we take a limit of vanishing shear viscosity.

## C.1 Navier-Stokes equation

In the following it is convenient to separate the velocity of the fluid into potential and vortical part

$$v_i = \partial_i \phi + v_i^\psi = \partial_i \phi + \partial_i^* \psi, \tag{107}$$

where $\phi$ is the velocity potential and $\psi$ is a stream function. The velocity potential is harmonic $\Delta \phi = 0$ as a consequence of incompressibility of the fluid. We rewrite the Navier-Stokes equation (7) as

$$\partial_t(\partial_i \phi + v_i^\psi) - v_i^* \omega = -\partial_i \left( \tilde{p} + \frac{v_j^2}{2} \right) + v_e \Delta v_i^\psi. \tag{108}$$

We integrate the $y$-component of this equation in $y$ from $-\infty$ to $h(x,t)$ and obtain

$$\partial_t \phi + \frac{1}{2}(\vec{\nabla}\phi + \vec{v}_\psi)^2 + \tilde{p} + \int_{-\infty}^h v_x \omega \, dy = -\int_{-\infty}^h (\partial_t v_y^\psi - v_e \Delta v_y^\psi) dy. \tag{109}$$

Let us now make the following assumptions in the limit $v_e \to 0$ (i) both potential and vortical components of velocity remain finite; (ii) the vortical component of velocity is essentially non-zero only in the narrow layer of thickness $\delta \sim \sqrt{v_e}$ near the boundary; (iii) the velocity of the fluid tangent to the boundary vanishes at the free surface; (iv) the vortical component of the velocity normal to the boundary vanishes.

The conditions (i) and (ii) mean that the vorticity might diverge as $1/\delta \sim v_e^{-1/2}$. Nevertheless, it is clear that the right hand side of (109) vanishes in the limit of $v_e \to 0$. However the term $\int_{-\infty}^h v_x \omega \, dy \approx \int_{-\infty}^h v_x \partial_x v_y \, dy$ does not vanish in this limit in the presence of odd viscosity and results in a non trivial contribution in the form of,

$$\int_{-\infty}^h v_x \partial_x v_y \, dy = \int_{-\infty}^h (v_x^\psi + \partial_x \phi) \partial_x (\partial_y \phi + v_y^\psi) \, dy \approx \int_{-\infty}^h (v_x^\psi + \partial_x \phi)(\partial_x \partial_y \phi) \, dy$$

$$\approx \int_{-\infty}^h (\partial_x \phi)(\partial_x \partial_y \phi) \, dy = \frac{1}{2}(\partial_x \phi)^2 \bigg|_{y=h}. \tag{110}$$

Using the conditions (iii) and (iv) in the left hand side of (109) we obtain the following approximate condition at the boundary $y = h(x,t)$

$$\partial_t \phi + \frac{1}{2}(\partial_x \phi)^2 + \frac{1}{2}(\partial_y \phi)^2 + \tilde{p} = 0. \tag{111}$$

In the remainder of this section we will derive an expression for $\tilde{p}$ in terms of the potential $\phi$ and justify some of our approximations.

Let us introduce a local coordinate system $(s,n)$ as shown in Fig. 1. In these curvilinear coordinates the tangential and normal boundary conditions are given by,

$$T_{nn} = -p + 2v_e \partial_n v_n + v_o(\partial_s v_n + \partial_n v_s - \kappa v_s) = 0, \tag{112}$$
$$T_{sn} = v_e(\partial_s v_n + \partial_n v_s - \kappa v_s) - 2v_o \partial_n v_n = 0, \tag{113}$$

where expressions for vorticity and incompressibility conditions are given by

$$\omega = \partial_n v_s - \partial_s v_n + \kappa v_s, \tag{114}$$
$$\partial_i v_i = \partial_n v_n + \partial_s v_s + \kappa v_n = 0. \tag{115}$$

## C.2 Tangent stress boundary condition

Using (114,115) we rewrite the tangent stress boundary condition (113) as

$$\partial_s v_s + \kappa v_n = -\frac{\nu_e}{\nu_o}\left(\partial_s v_n - \kappa v_s + \frac{1}{2}\omega\right),\tag{116}$$

Taking a limit $\nu_e \to 0$ and assuming that the divergence of $\omega \sim \nu_e^{-1/2}$ we obtain

$$\partial_s v_s + \kappa v_n = O(\sqrt{\nu_e}) \to 0.\tag{117}$$

This condition is consistent with the previously used assumption that $v_s = 0$ in linear approximation when $\nu_e \to 0$. We will use the fact that $v_s$ is of at least the second order in the amplitude of the wave.

## C.3 Kinematic boundary condition

The kinematic boundary condition (9) can be rewritten as

$$h_t = v_n = \partial_n \phi + v_n^\psi.\tag{118}$$

Neglecting $v_n^\psi$ and using $\partial_n \phi \approx \partial_y \phi - h_x \partial_x \phi$ we obtain

$$h_t = \partial_y \phi - h_x \partial_x \phi\tag{119}$$

as a kinematic boundary condition.

## C.4 Normal stress boundary condition

Let us now rewrite the boundary condition for the normal part of the stress tensor as

$$T_{nn} = -\tilde{p} + 2\nu_o(\partial_s v_n - \kappa v_s) + 2\nu_e \partial_n v_n = 0.\tag{120}$$

Here we introduced $\tilde{p} = p - \nu_o \omega$ and used (114). We immediately obtain for the boundary value of the modified pressure

$$\tilde{p} = 2\nu_o(\partial_s v_n - \kappa v_s) + 2\nu_e \partial_n v_n.\tag{121}$$

Taking a limit $\nu_e \to 0$ and ignoring $\kappa v_s$ term[9] the higher order terms in this expression we obtain

$$\tilde{p} \approx 2\nu_o \partial_s v_n,\tag{122}$$

and we can use it in (111) to obtain

$$\partial_t \phi + \frac{1}{2}(\partial_x \phi)^2 + \frac{1}{2}(\partial_y \phi)^2 + 2\nu_o \partial_s v_n = 0.\tag{123}$$

We proceed neglecting all terms smaller than quadratic ones as

$$\partial_s v_n = (-\partial_x - h_x \partial_y)v_n = -\partial_x v_n - h_x \partial_y v_y = -\partial_x v_n + h_x \partial_x v_x = -\partial_x v_n = -\partial_x(\partial_y \phi - h_x \partial_x \phi).\tag{124}$$

Here we again used the fact that $v_s$ and $v_x$ are quadratic in the amplitude. Using this relation we can transform (123) to

$$\partial_t \phi + \frac{1}{2}(\partial_x \phi)^2 + \frac{1}{2}(\partial_y \phi)^2 = 2\nu_o \partial_x(\partial_y \phi - h_x \partial_x \phi).\tag{125}$$

This is the equation that we derived in Sec. 5 Eq. 44. This equation can be expressed in terms of $\tilde{\phi}(x,t) = \phi(x, h(x,t), t)$ leading to the non-linear Hamiltonian system defined in Eqs. 49 and 50.

---

[9] The tangent velocity in the limit $\nu_e \to 0$ is of the second order of smallness in the amplitude of the wave.

# D Nonlinear periodic chiral surface waves in chiral Burgers equation

The goal of this appendix is to derive a periodic moving wave solution of the chiral Burgers equation (66) used in Section 7. We look for a moving wave solution of the form $u = u(x-Ut)$. We substitute it into (66) and obtain:

$$\partial_x\left(-Uu + u^2 - 2i\,\nu_o u_x\right) = 0. \tag{126}$$

Integrating the above equation and setting the overall constant as $A$ we obtain,

$$-Uu + u^2 - 2i\,\nu_o u_x = A. \tag{127}$$

Integrating this equation gives:

$$\frac{2i\,\nu_o}{U_1 - U_2}\log\left(\frac{u - U_1}{u - U_2}\right) = x - x_0 - Ut - ia. \tag{128}$$

Here $U_{1,2} = \frac{U}{2}(1 \pm \sqrt{1 + 4A/U^2})$ and $a$ is a real constant of integration. In the following we set $x_0 = 0$ without loss of generality. Solving with respect to $u$ we have

$$u = \frac{U_1 - U_2 e^{ik(x-Ut)+ka}}{1 - e^{ik(x-Ut)+ka}}, \tag{129}$$

where $k = -\frac{1}{2\nu_o}(U_1 - U_2)$ and from analyticity of $u$ in the lower half plane we fix $a > 0$. Fixing $k > 0$, analytically continuing $x \to z = x + iy$ and requiring that $u \to 0$ as $y \to -\infty$ we obtain $U_2 = 0$. This means that integration constant $A = 0$ and $U_1 = U$. We obtain the solution

$$u = \frac{U}{1 - e^{ik(x-Ut)+ka}}, \tag{130}$$

with $k > 0$, $a > 0$ and

$$U = -2\nu_o k, \tag{131}$$

for $u$ consistent with boundary conditions and analyticity requirements. The overall velocity field $\tilde{u} = u + \bar{u}$ is then given by

$$\tilde{u} = -2\nu_o k\left(1 - \frac{\sinh(ka)}{\cosh(ka) - \cos(k(x-Ut))}\right). \tag{132}$$

In the limit of $a \to +\infty$ while keeping $k > 0$ fixed, we obtain

$$\tilde{u} = -2\nu_o k(1 - \tanh(ka)) + 2\nu_o k\frac{\tanh(ka)}{\cosh(ka)}\cos(k(x-Ut)). \tag{133}$$

The first term vanishes as $a \to +\infty$ for $k > 0$. In this limit we recover the linearized Lamb's solution,

$$\tilde{u} \approx 4\nu_o k e^{-ka}\cos(k(x-Ut)). \tag{134}$$

We can now compute the non linear profile of the height function $h(x,t)$ using the kinematic condition of the form $h = \int^t \phi_y\,dt = -\int^t \tilde{u}^H dt$. We have defined $\tilde{u}^H = i(\bar{u} - u)$ and using the expression for $u$ in Eq. 67 we get

$$\tilde{u}^H = -2\nu_o k\frac{\sin(k(x + 2\nu_o kt))}{\cos(k(x + 2\nu_o kt)) - \cosh(ka)}. \tag{135}$$

Using the above expression we can obtain $h(x,t)$ as

$$h(x,t) = -\int^t \tilde{u}^H dt = \frac{1}{k}\log[\cosh(ak) - \cos(k(x+2\nu_o kt))] + \text{const.} \quad (136)$$

The above expression was restricted to $k > 0$. We can now recover the linearized Lamb limit by taking $a \to +\infty$. In this limit up to an overall constant we obtain,

$$h(x,t) \approx -2\frac{e^{-ak}}{k}\cos\left(k(x+2\nu_o|k|t)\right). \quad (137)$$

The above expression is consistent with the linearized Lamb solutions as presented in the main text.

# E   Multi-pole solutions for chiral Burgers equation

In this section, we derive the multi-pole solution for the non-linear chiral Burgers equation,

$$u_t + 2uu_x - 2i\nu_o u_{xx} = 0, \quad (138)$$

where we additionally require that $u(x,t)$ is analytic in the lower half of complex plane after analytic continuation in $x$. We make the following ansatz,

$$u(x,t) = -2i\nu_o \sum_{j=1}^{N}\frac{1}{x-z_j}, \quad (139)$$

which is analytic in the lower half plane and all the poles $z_j$ are in the upper half plane. Substituting the above ansatz in the equation, we obtain a dynamical system for the poles,

$$\dot{z}_j = 4i\nu_o \sum_{k\neq j}^{N}\frac{1}{z_k - z_j}. \quad (140)$$

The above dynamical system corresponds to a valid solution of Eq. 138 only for a finite time as at least one of the poles inevitably hits the real axis after which the the physical description through the chiral Burgers equations becomes untenable. In other words, beyond this time scale either higher order corrections become important or the boundary layer approximation breaks down. However, not all multipole solutions are unstable. In particular, we can arrange the poles such that it sums up to the non-linear periodic form given in Eq. 130. Consider $z_j(t) = 2\pi j/k$ spaced evenly at distance $2\pi/k$. For such an arrangement of poles, the multipole ansatz becomes,

$$u(x,t) = -\nu_o k - \frac{2i\nu_o}{x-Ut-ia} - \sum_{n=1}^{\infty}\left(\frac{2i\nu_o}{(x-Ut-ia)-\frac{2n\pi}{k}} + \frac{2i\nu_o}{(x-Ut-ia)+\frac{2n\pi}{k}}\right). \quad (141)$$

The constant first term is to satisfy the boundary condition $u \to 0$ as $z \to \infty$. The terms in the above expression sum to,

$$u(x,t) = -\nu_o k\left(1 + i\cot\left(\frac{k(x-Ut-ia)}{2}\right)\right). \quad (142)$$

The above expression can be recast in a more familiar form that reproduces equation 130,

$$u(x,t) = \frac{-2\nu_o k}{1 - e^{ik(x-Ut)+ka}}. \quad (143)$$

The velocity $U = -2\nu_o k$ can be determined by regularizing the summation in Eq. 140. In next few sections we consider single and double pole solutions.

## E.1 Single-pole solution

In a single pole solution $N = 1$ the system (140) degenerates to $z_1 = const$ and we have

$$u(x,t) = -2i v_o \sum_{j=1}^{N} \frac{1}{x - ia} \tag{144}$$

to be a solution of (138). This solution is time-independent for $u(x,t)$ and corresponds to the real $\tilde{u} = u + \bar{u}$ given by

$$\tilde{u} = -\frac{4 v_o a}{x^2 + a^2} . \tag{145}$$

The profile of the wave $h(x,t)$ corresponding to this solution can be found as

$$h = \frac{4 v_o x}{x^2 + a^2} t \tag{146}$$

and grows in time linearly. In time of the order of $t_* = a^2 / v_o$ the curvature of the profile becomes significant and the assumptions of small curvature used in deriving (66) are violated.

## E.2 Two-pole solution

Let us consider an example of two poles. We have

$$\dot{z}_{1,2}(t) = \mp \frac{4i v_o}{z_1 - z_2} . \tag{147}$$

These equations are easy to solve to obtain

$$z_{1,2}(t) = z_0 \pm \sqrt{-4i v_o (t - t_0) + C} , \tag{148}$$

where $\text{Im}(z_0) > 0$ and $C$ is an arbitrary real constant. It is clear that at some time one of the poles approaches the real axis. At this point the assumptions of small nonlinearity used in the derivation of chiral Burgers equation are violated. A typical time scale for this is $t_* = (\text{Im}(z_0))^2 / v_o$.

## F General family of non-linear equations

In this section, we put together a one parameter family of non-linear equations that contains both Benjamin-Davis-One (BDO) and chiral Burgers equation as a special cases. We introduce the following non-linear equation,

$$u_t + 2\lambda u u_x - 2(1 - \lambda) u^H u_x^H + 2 v_o u_{xx}^H = 0. \tag{149}$$

In the above family of non-linear equations, we obtain BDO for $\lambda = 1$ and chiral Burgers type equation for $\lambda = 1/2$. In the limiting case of $\lambda = 0$, we obtain a new non-linear equation which possesses multi-soliton solutions,

$$u_t - 2 u^H u_x^H + 2 v_o u_{xx}^H = 0. \tag{150}$$

However, unlike the BDO case where multi-soliton solutions are scattering states, the multi-soliton solutions of Eq. 150 form bound states. Generally, whether these solitons are of bound

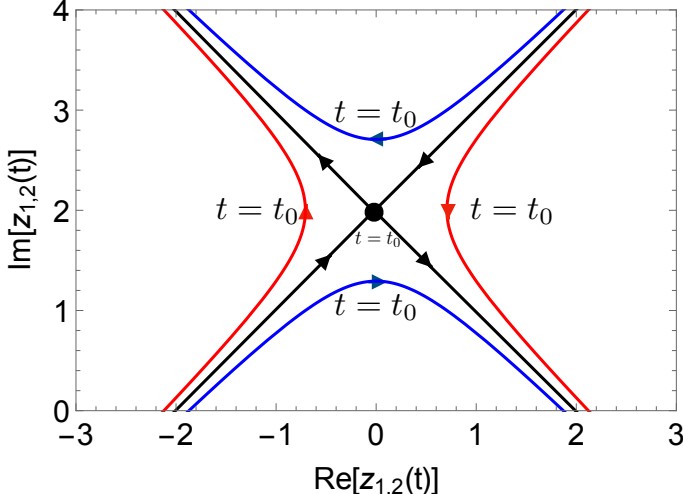

Figure 7: Dynamics of the two poles given by Eq.148 in the complex plane with time as the parameter. We have set $v_o = 1$, $t_0 = 0$ and $z_0 = i$. The three colors correspond to $C = -0.5$ (Blue), $C = 0.5$(Red) and $C = 0$ (Black).

state type or scattering type is dictated by the parameter $\lambda$. Studying the multi-soliton dynamics of this general family sheds light on the dynamics of the chiral Burgers case which contains the physics of non-linear odd surface waves discussed in this work. One can write down a multi-pole solution for the equation 149 as,

$$u(x,t) = -2i\,v_o \sum_{j=1}^{N} \frac{1}{x - z_j} + c.c. \tag{151}$$

Substituting the multi-pole solution in Eq. 149, we obtain the dynamics of poles as,

$$\dot{z}_j = 4i\,v_o \left( \sum_{k=1, k \neq j}^{N} \frac{1}{z_k - z_j} + (1 - 2\lambda) \sum_{k=1}^{N} \frac{1}{\bar{z}_k - z_j} \right). \tag{152}$$

The above dynamical system corresponds to the celebrated integrable BDO equation for $\lambda = 1$. The pole dynamics for the BDO corresponds to the scattering states of multi-soliton solutions. For $\lambda = 0$ we have a new kind of multi-soliton solution corresponding to Eq. 150 which form bound states of the multi-soliton solutions. The stability of these bound states is an interesting future direction. For $\lambda = 1/2$, we recover chiral Burgers equation where the second term in the pole dynamics drops out and we obtain multi-pole dynamics of Eq. 140.

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
