# Peer review of "Odd surface waves in two-dimensional incompressible fluids"

_SciPost Physics, doi:SciPost Phys. 5, 010 (2018)_

## Round 3 · Referee Report · Anonymous (Referee 1) · 2018-4-12

Strengths

1) First study addressing the effect of odd viscosity on surface gravity waves, from linear to nonlinear dynamics. [] 2) Paper well organized

Weaknesses

1) The effect of gravity seems to be discussed only in the part on linear waves. [] 2) It is unclear at this stage how such odd viscous surface waves could be observed, but this is an interesting perspective!

Report

The authors revisit the problem of viscous deep water gravity waves. The novelty is to add an odd viscosity term in the case of a two-dimensional vertical slice of the fluid. In a first part, the authors derive properties of the linearized dynamics, and discuss an interesting physical consequence of adding odd viscosity on the wave structure: odd viscous terms lead to a Prandtl-like boundary layer. In a second part, the authors derive non-linear wave equations, and find a complex Burger equation. This equation had been previously studied, but without underlying physical model. Odd viscosity thus makes possible a physical realization of this model.

$$] I think this is a very interesting work. Odd viscous waves and non-linear surface gravity waves are two active research fields, and this paper is a welcome contribution to bridge the gap between them. One may wonder about actual physical realization of the problem addressed in the paper, but the study presents a clear interest on its own, albeit academic at this stage. The paper is clearly written, with an effort to separate physical discussions in the main text and technical computations in several appendices. The results are also put in perspective with respect to previous work on the subject. For these reasons I support publication in SciPost Physics. Please find below some comments and questions. [$$
1) I found very interesting that adding odd viscosity $\nu_o$ changes the scaling of vorticity with shear viscosity $\nu_e$. The scaling of of layer thickness as $\nu_e^{1/2}$ and of vorticity as $\nu_e^{-1/2}$ is reminiscent of the Prandtl boundary layer in the presence of a solid wall with no-slip condition, which is consistent with the observation that odd-viscous terms induces an effective condition of no tangential flow at the free surface. The (nonlinear) Prandtl equation are known to develop finite time singularities, and in actual flows this boundary layer can be detached from the wall above a critical Reynolds number. Could similar singularities and boundary layer detachment happens in this setting?
$$] 2) page 13. Is the Hamiltonian structure surprising because it is present in the 1D equation at the end of the derivation, starting from a model that included shear viscosity? As far as the initial flow model is concerned, does the system with odd viscosity and without shear viscosity has also an Hamiltonian formulation, and does it bear similarities with the Hamiltonian formulation of 2D (horizontal) rotating shallow water flows, given the similarities between Coriolis force in that case and odd viscous terms? For instance, in the Hamiltonian formulation of rotating shallow water equations, rotation does not enter into the Hamiltonian, but enter into the Poisson tensor. [$$
3) A question on the structure of the paper. I am a bit confused by the role of gravity. In the first part, the authors discuss the combined effect of odd viscosity, shear viscosity and gravity. But as far as I understand it, gravity is absent in the second part of the paper dealing with nonlinearities. What is the effect of adding back corrections due to the presence of gravity into the non-linear problem? Or what is the interest of considering gravity in the first place?
$$] 4) Could such odd deep surface waves could be implemented in actual experiments? Will the theoretical results be robust to 3D perturbations? [$$
5) Details:
$$] -The term 'odd surface wave' could be also used to describe 'odd shallow water wave' in horizontal 2D flow models with odd-viscosity. The waves described in this paper and in the title could be dubbed 'odd deep water surface waves'. Similarly, the name of H. Lamb is associated with a variety of waves in different contexts. To avoid confusions when speaking about Lamb solution, the authors could perhaps use the term 'viscous Lamb surface waves'. [$$
-page 2: is has $\rightarrow$ has
$$] -page 3: background metric $\rightarrow$ please add reference. [$$
-page 3: 2D ocean $\rightarrow$ a 2D vertical slice of an ocean
$$] -page 11: how do we conclude from $\partial_x v_x=0 $ that $v_x=0$ (or second order in amplitude) ? [$$
-page 12 note 4: I am confused by 'the difference between $\phi$ and $\tilde{\phi}$'. How to compare them if one is a function of $(x,t)$ and the other a function of $(x,y,t)$ ?
$$] -page 23: please explain your notation $i0$ [$$
-page 24: equation just above Eq. (99) has many typos

Requested changes

Please see the 6 minor comments above.

  • validity: high
  • significance: high
  • originality: good
  • clarity: high
  • formatting: excellent
  • grammar: excellent

Anonymous on 2018-06-14  [id 274]

(in reply to Report 1 on 2018-04-12)

Response to the referee comments and list of changes.

Referee: The authors revisit the problem of viscous deep water gravity waves. The novelty is to add an odd viscosity term in the case of a two-dimensional vertical slice of the fluid. In a first part, the authors derive properties of the linearized dynamics, and discuss an interesting physical consequence of adding odd viscosity on the wave structure: odd viscous terms lead to a Prandtl-like boundary layer. In a second part, the authors derive non-linear wave equations, and find a complex Burger equation. This equation had been previously studied, but without underlying physical model. Odd viscosity thus makes possible a physical realization of this model. 

I think this is a very interesting work. Odd viscous waves and non-linear surface gravity waves are two active research fields, and this paper is a welcome contribution to bridge the gap between them. One may wonder about actual physical realization of the problem addressed in the paper, but the study presents a clear interest on its own, albeit academic at this stage. The paper is clearly written, with an effort to separate physical discussions in the main text and technical computations in several appendices. The results are also put in perspective with respect to previous work on the subject. For these reasons I support publication in SciPost Physics. Please find below some comments and questions. 

Authors: We thank the referee for reviewing our work and recommending it for publication to SciPost Physics. Below we respond to the comments and questions of the referee.

Referee: I found very interesting that adding odd viscosity $\nu_o$ changes the scaling of vorticity with shear viscosity $\nu_e$ . The scaling of layer thickness as $\sqrt{\nu_e}$ and of vorticity as $\nu_e^{-1/2}$ is reminiscent of the Prandtl boundary layer in the presence of a solid wall with no-slip condition, which is consistent with the observation that odd-viscous terms induces an effective condition of no tangential flow at the free surface. The (nonlinear) Prandtl equation are known to develop finite time singularities, and in actual flows this boundary layer can be detached from the wall above a critical Reynolds number. Could similar singularities and boundary layer detachment happens in this setting? AUTHORS: The boundary layer considered in this work is specific to the no-stress boundary condition that is often used in the case of two-fluid interfaces. For the case of no-slip or flow dependent boundary condition, two of us have shown in a previous publication that the odd viscosity completely drops out of the equations and does not modify the velocity field when the fluid is incompressible with constant density (see ref. [30]). In other words, the odd viscosity with no slip boundary conditions does not generate a boundary layer dynamically. There is still a possibility that due to microscopic reasons the no-slip boundary condition itself is not the correct one. We do not have anything to say about this possibility at the moment.

This result motived us to study dynamical surfaces with no-stress boundary conditions where the physics of the boundary layer can be introduced. We do see finite time singularities in the small angle approximation of the weakly non-linear regime. Most of the generic soliton configurations run into finite time singularities. These finite time singularities are also present in an ideal inviscid fluid with no gravity (see ref. [42]). We do not know how this finite time singularity alters the boundary layer, since it appears in the more general dynamical system (beyond small angle calculations) defined in section 5 and 6. It may be possible that the vortices are shed into the bulk in response to the singularities. A full simulation with a dynamical boundary is required to elucidate this phenomena. This is the subject of a future research direction.

Referee: 2) page 13. Is the Hamiltonian structure surprising because it is present in the 1D equation at the end of the derivation, starting from a model that included shear viscosity? As far as the initial flow model is concerned, does the system with odd viscosity and without shear viscosity has also an Hamiltonian formulation, and does it bear similarities with the Hamiltonian formulation of 2D (horizontal) rotating shallow water flows, given the similarities between Coriolis force in that case and odd viscous terms? For instance, in the Hamiltonian formulation of rotating shallow water equations, rotation does not enter into the Hamiltonian, but enter into the Poisson tensor.

AUTHORS: The existence of a Hamiltonian structure is somewhat surprising but is very natural as well, as clarified by the referee’s comment. On the one hand, it is known that the limit of vanishing shear viscosity is very special. The dissipation is not necessarily zero in this limit (e.g., in turbulence) - a phenomenon known as the “dissipative anomaly”. However, a simple estimate of the dissipation in the boundary layer using the scales derived in our paper shows that the dissipation is vanishing in case of surface waves. This makes the Hamiltonian structure very natural. We also agree with the referee’s analogy with rotating shallow water waves. In fact, one can think about odd viscosity as higher gradient corrections to Coriolis forces, when the Coriolis forces themselves are vanishing (see Ref. 26 Sec 3).

Referee: 3) A question on the structure of the paper. I am a bit confused by the role of gravity. In the first part, the authors discuss the combined effect of odd viscosity, shear viscosity and gravity. But as far as I understand it, gravity is absent in the second part of the paper dealing with nonlinearities. What is the effect of adding back corrections due to the presence of gravity into the non-linear problem? Or what is the interest of considering gravity in the first place?

AUTHORS: There are a couple of reasons we started with gravity in equations. First, we wanted to highlight that the non-dissipative dispersion terms come from both gravity and odd viscosity. The shear viscosity is a small dissipative correction. Without odd viscosity, the gravity term is the only non-dissipative term and switching it off kills the surface dynamics. However, with odd viscosity one can generate dispersive dynamics in the absence of gravity.

Another interesting point is that the surface gravity waves are not chiral and have both left and right movers. This can be seen directly from the dispersion since there are both positive and negative branches (see Fig. 2 ). In the limit of zero gravity, one of these branches becomes a zero mode and the other becomes a unidirectional chiral branch. In section 5, 7 we wanted to highlight the chiral dynamics of the surface waves when odd viscosity is the dominant term. We re-introduced gravity in section 6 of the paper, showing how the nonlinear equations can be written in the presence of gravity. Keeping both odd viscosity and gravity on the same footing takes us into an interesting dynamical regime, but it significantly complicates the analysis of the equations. This is an interesting problem for future research.

Referee: 4) Could such odd deep surface waves could be implemented in actual experiments? Will the theoretical results be robust to 3D perturbations?

AUTHORS: There are a few candidate experiments where this phenomenon can be seen. For example, chiral active fluids and mechanical systems that break parity at an intrinsic level. However, it is not clear if odd viscosity will be a dominant term in the dynamical equations describing these fluids. The odd viscosity is not the only parity breaking term in 2D systems with isotropy. In particular, it is known that the rotational viscosity (aka “odd stress”) plays a big role in actual experiments. Such experimental work is being carried out, e.g., in William Irvine’s lab with magnetic colloids and will be soon published.

3D perturbations open up a whole new direction in this field. In 3D, parity breaking is not compatible with isotropy. What this means for 3D perturbations to 2D odd viscosity flow is that the out-of-plane dynamics will be governed by different equations than the in-plane dynamics. Furthermore there are vortex stretching terms that are possible in 3D that will alter the boundary layer structure. These effects are indeed relevant and further study should be specialized to the experimental setup at hand. We feel that even these speculations are outside the scope of the present work, as there exist no previous studies on which to build our intuition.

Referee: 5) Details -The term 'odd surface wave' could be also used to describe 'odd shallow water wave' in horizontal 2D flow models with odd-viscosity. The waves described in this paper and in the title could be dubbed 'odd deep water surface waves'. Similarly, the name of H. Lamb is associated with a variety of waves in different contexts. To avoid confusions when speaking about Lamb solution, the authors could perhaps use the term 'viscous Lamb surface waves’.

AUTHORS: We would like to emphasize that our system is not limited to water waves. We have clarified the above concern in the main draft through the following paragraph in the introduction.

“In this work, we consider the classical problem of deep water surface waves with the addition of odd viscosity. For simplicity, we refer to these as simply {\it odd surface waves}. We do not, however, consider the odd version of shallow water surface waves, which we reserve for future investigations.”

We approach the “Lamb’s solution” issue in the same way. Explicitly, we say that we consider the viscous Lamb surface wave solution, and that for simplicity, we refer to this as Lamb’s solution later on in the paper.

Response to specific comments:

-page 2: is has → has

-page 3: background metric → please add reference.

Authors: Added reference.

-page 3: 2D ocean → a 2D vertical slice of an ocean

Authors: We replaced “2D ocean” with “2D fluid”.

-page 11: how do we conclude from $\partial_x v_x=0$ that v_x=0 (or second order in amplitude) ?

Authors: From our scaling analysis of linearized odd surface wave solutions, we obtain that $v_x \sim 0$ as $\nu_e \rightarrow 0$. Thus in the $\nu_e \rightarrow 0$ limit, $v_x$ must be at least second order in amplitude. The details of this scaling analysis is given in Sec 4.2 and in Appendix C.

-page 12 note 4: I am confused by 'the difference between $\phi$ and $\tilde \phi$. How to compare them if one is a function of (x,t) and the other a function of (x, y, t)?

Authors: $\tilde \phi(x,t)=\phi(x, h(x,t),t)$ where y is taken at the dynamical boundary $h(x,t)$.

-page 23: please explain your notation i0

Authors: i0 is equivalent to $i\epsilon$ with $\epsilon\rightarrow +0$. (added to the manuscript)

-page 24: equation just above Eq. (99) has many typos

Authors: Typos fixed.

---

## Round 4 · Author Response

Response to the referee comments and list of changes.

Referee: The authors revisit the problem of viscous deep water gravity waves. The novelty is to add an odd viscosity term in the case of a two-dimensional vertical slice of the fluid. In a first part, the authors derive properties of the linearized dynamics, and discuss an interesting physical consequence of adding odd viscosity on the wave structure: odd viscous terms lead to a Prandtl-like boundary layer. In a second part, the authors derive non-linear wave equations, and find a complex Burger equation. This equation had been previously studied, but without underlying physical model. Odd viscosity thus makes possible a physical realization of this model. 

I think this is a very interesting work. Odd viscous waves and non-linear surface gravity waves are two active research fields, and this paper is a welcome contribution to bridge the gap between them. One may wonder about actual physical realization of the problem addressed in the paper, but the study presents a clear interest on its own, albeit academic at this stage. The paper is clearly written, with an effort to separate physical discussions in the main text and technical computations in several appendices. The results are also put in perspective with respect to previous work on the subject. For these reasons I support publication in SciPost Physics. Please find below some comments and questions. 

Authors: We thank the referee for reviewing our work and recommending it for publication to SciPost Physics. Below we respond to the comments and questions of the referee.

Referee: I found very interesting that adding odd viscosity $\nu_o$ changes the scaling of vorticity with shear viscosity $\nu_e$ . The scaling of layer thickness as $\sqrt{\nu_e}$ and of vorticity as $\nu_e^{-1/2}$ is reminiscent of the Prandtl boundary layer in the presence of a solid wall with no-slip condition, which is consistent with the observation that odd-viscous terms induces an effective condition of no tangential flow at the free surface. The (nonlinear) Prandtl equation are known to develop finite time singularities, and in actual flows this boundary layer can be detached from the wall above a critical Reynolds number. Could similar singularities and boundary layer detachment happens in this setting? AUTHORS: The boundary layer considered in this work is specific to the no-stress boundary condition that is often used in the case of two-fluid interfaces. For the case of no-slip or flow dependent boundary condition, two of us have shown in a previous publication that the odd viscosity completely drops out of the equations and does not modify the velocity field when the fluid is incompressible with constant density (see ref. [30]). In other words, the odd viscosity with no slip boundary conditions does not generate a boundary layer dynamically. There is still a possibility that due to microscopic reasons the no-slip boundary condition itself is not the correct one. We do not have anything to say about this possibility at the moment.

This result motived us to study dynamical surfaces with no-stress boundary conditions where the physics of the boundary layer can be introduced. We do see finite time singularities in the small angle approximation of the weakly non-linear regime. Most of the generic soliton configurations run into finite time singularities. These finite time singularities are also present in an ideal inviscid fluid with no gravity (see ref. [42]). We do not know how this finite time singularity alters the boundary layer, since it appears in the more general dynamical system (beyond small angle calculations) defined in section 5 and 6. It may be possible that the vortices are shed into the bulk in response to the singularities. A full simulation with a dynamical boundary is required to elucidate this phenomena. This is the subject of a future research direction.

Referee: 2) page 13. Is the Hamiltonian structure surprising because it is present in the 1D equation at the end of the derivation, starting from a model that included shear viscosity? As far as the initial flow model is concerned, does the system with odd viscosity and without shear viscosity has also an Hamiltonian formulation, and does it bear similarities with the Hamiltonian formulation of 2D (horizontal) rotating shallow water flows, given the similarities between Coriolis force in that case and odd viscous terms? For instance, in the Hamiltonian formulation of rotating shallow water equations, rotation does not enter into the Hamiltonian, but enter into the Poisson tensor.

AUTHORS: The existence of a Hamiltonian structure is somewhat surprising but is very natural as well, as clarified by the referee’s comment. On the one hand, it is known that the limit of vanishing shear viscosity is very special. The dissipation is not necessarily zero in this limit (e.g., in turbulence) - a phenomenon known as the “dissipative anomaly”. However, a simple estimate of the dissipation in the boundary layer using the scales derived in our paper shows that the dissipation is vanishing in case of surface waves. This makes the Hamiltonian structure very natural. We also agree with the referee’s analogy with rotating shallow water waves. In fact, one can think about odd viscosity as higher gradient corrections to Coriolis forces, when the Coriolis forces themselves are vanishing (see Ref. 26 Sec 3).

Referee: 3) A question on the structure of the paper. I am a bit confused by the role of gravity. In the first part, the authors discuss the combined effect of odd viscosity, shear viscosity and gravity. But as far as I understand it, gravity is absent in the second part of the paper dealing with nonlinearities. What is the effect of adding back corrections due to the presence of gravity into the non-linear problem? Or what is the interest of considering gravity in the first place?

AUTHORS: There are a couple of reasons we started with gravity in equations. First, we wanted to highlight that the non-dissipative dispersion terms come from both gravity and odd viscosity. The shear viscosity is a small dissipative correction. Without odd viscosity, the gravity term is the only non-dissipative term and switching it off kills the surface dynamics. However, with odd viscosity one can generate dispersive dynamics in the absence of gravity.

Another interesting point is that the surface gravity waves are not chiral and have both left and right movers. This can be seen directly from the dispersion since there are both positive and negative branches (see Fig. 2 ). In the limit of zero gravity, one of these branches becomes a zero mode and the other becomes a unidirectional chiral branch. In section 5, 7 we wanted to highlight the chiral dynamics of the surface waves when odd viscosity is the dominant term. We re-introduced gravity in section 6 of the paper, showing how the nonlinear equations can be written in the presence of gravity. Keeping both odd viscosity and gravity on the same footing takes us into an interesting dynamical regime, but it significantly complicates the analysis of the equations. This is an interesting problem for future research.

Referee: 4) Could such odd deep surface waves could be implemented in actual experiments? Will the theoretical results be robust to 3D perturbations?

AUTHORS: There are a few candidate experiments where this phenomenon can be seen. For example, chiral active fluids and mechanical systems that break parity at an intrinsic level. However, it is not clear if odd viscosity will be a dominant term in the dynamical equations describing these fluids. The odd viscosity is not the only parity breaking term in 2D systems with isotropy. In particular, it is known that the rotational viscosity (aka “odd stress”) plays a big role in actual experiments. Such experimental work is being carried out, e.g., in William Irvine’s lab with magnetic colloids and will be soon published.

3D perturbations open up a whole new direction in this field. In 3D, parity breaking is not compatible with isotropy. What this means for 3D perturbations to 2D odd viscosity flow is that the out-of-plane dynamics will be governed by different equations than the in-plane dynamics. Furthermore there are vortex stretching terms that are possible in 3D that will alter the boundary layer structure. These effects are indeed relevant and further study should be specialized to the experimental setup at hand. We feel that even these speculations are outside the scope of the present work, as there exist no previous studies on which to build our intuition.

Referee: 5) Details -The term 'odd surface wave' could be also used to describe 'odd shallow water wave' in horizontal 2D flow models with odd-viscosity. The waves described in this paper and in the title could be dubbed 'odd deep water surface waves'. Similarly, the name of H. Lamb is associated with a variety of waves in different contexts. To avoid confusions when speaking about Lamb solution, the authors could perhaps use the term 'viscous Lamb surface waves’.

AUTHORS: We would like to emphasize that our system is not limited to water waves. We have clarified the above concern in the main draft through the following paragraph in the introduction.

“In this work, we consider the classical problem of deep water surface waves with the addition of odd viscosity. For simplicity, we refer to these as simply {\it odd surface waves}. We do not, however, consider the odd version of shallow water surface waves, which we reserve for future investigations.”

We approach the “Lamb’s solution” issue in the same way. Explicitly, we say that we consider the viscous Lamb surface wave solution, and that for simplicity, we refer to this as Lamb’s solution later on in the paper.

---

## Round 4 · List of Changes

Response to specific comments and list of changes:

-page 2: is has → has

-page 3: background metric → please add reference.

Authors: Added reference.

-page 3: 2D ocean → a 2D vertical slice of an ocean

Authors: We replaced “2D ocean” with “2D fluid”.

-page 11: how do we conclude from $\partial_x v_x=0$ that v_x=0 (or second order in amplitude) ?

Authors: From our scaling analysis of linearized odd surface wave solutions, we obtain that $v_x \sim 0$ as $\nu_e \rightarrow 0$. Thus in the $\nu_e \rightarrow 0$ limit, $v_x$ must be at least second order in amplitude. The details of this scaling analysis is given in Sec 4.2 and in Appendix C.

-page 12 note 4: I am confused by 'the difference between $\phi$ and $\tilde \phi$. How to compare them if one is a function of
(x,t) and the other a function of (x, y, t)?

Authors: $\tilde \phi(x,t)=\phi(x, h(x,t),t)$ where y is taken at the dynamical boundary $h(x,t)$.

-page 23: please explain your notation i0

Authors: i0 is equivalent to $i\epsilon$ with $\epsilon\rightarrow +0$. (added to the manuscript)

-page 24: equation just above Eq. (99) has many typos

Authors: Typos fixed.

You are currently on this page

Resubmission 1801.10150v4 on 19 June 2018

---

## Editorial Decision

published